# THE SAMPLE COMPLEXITY OF ONLINE REINFORCEMENT LEARNING: A MULTI-MODEL PERSPECTIVE

**Michael Muehlebach**
Max Planck Institute for Intelligent Systems
Tuebingen, Germany
`michaelm@tuebingen.mpg.de`

**Zhiyu He**
Max Planck Institute for Intelligent Systems
Tuebingen, Germany
`zhiyu.he@tuebingen.mpg.de`

**Michael I. Jordan**
Inria Paris, France
University of California, Berkeley, USA
`jordan@cs.berkeley.edu`

## ABSTRACT

We study the sample complexity of online reinforcement learning in the general non-episodic setting of nonlinear dynamical systems with continuous state and action spaces. Our analysis accommodates a large class of dynamical systems ranging from a finite set of nonlinear candidate models to models with bounded and Lipschitz continuous dynamics, to systems that are parametrized by a compact and real-valued set of parameters. In the most general setting, our algorithm achieves a policy regret of $\mathcal{O}(N\epsilon^2 + d_{\mathrm{u}}\ln(m(\epsilon))/\epsilon^2)$, where $N$ is the time horizon, $\epsilon$ is a user-specified discretization width, $d_{\mathrm{u}}$ the input dimension, and $m(\epsilon)$ measures the complexity of the function class under consideration via its packing number. In the special case where the dynamics are parametrized by a compact and real-valued set of parameters (such as neural networks, transformers, etc.), we prove a policy regret of $\mathcal{O}(\sqrt{d_{\mathrm{u}}Np})$, where $p$ denotes the number of parameters, recovering earlier sample-complexity results that were derived for *linear time-invariant* dynamical systems. While this article focuses on characterizing sample complexity, the proposed algorithms are likely to be useful in practice, due to their simplicity, their ability to incorporate prior knowledge, and their benign transient behaviors.

## 1 INTRODUCTION

Reinforcement learning describes the situation where a decision-maker chooses actions to control a dynamical system, which is unknown a priori, to optimize a performance measure. At the core of reinforcement learning is the fundamental dilemma between choosing actions that reveal information about the dynamics and choosing actions that optimize performance. These are typically conflicting goals. We consider an online non-episodic setting, where the decision-maker is required to learn continuously and is unable to reset the state of the dynamical system. This further introduces the challenge that the information received by the learner is correlated over time and hence, standard statistical tools cannot be applied directly. Despite these important challenges, we provide a suite of online reinforcement learning algorithms that build on Hedge-type updates (Cesa-Bianchi & Lugosi, 2006) and posterior sampling reinforcement learning (Osband et al., 2013), while being straightforward to analyze, practically and theoretically relevant, and offering strong non-episodic, nonasymptotic, frequentist policy-regret guarantees that apply to continuous state-action systems. The algorithms sample from a posterior over the different model candidates (or an approximation thereof), apply the corresponding "certainty-equivalent" policies (Mania et al., 2019), while carefully introducing enough excitation to ensure that the posterior distribution over models converges sufficiently rapidly.

We consider three different settings. In the first setting, the decision-maker has access to a finite set of nonlinear candidate models that potentially describe the system dynamics (continuous state and action spaces). This setting is relevant for many practical engineering applications, where the choice of candidate models provides a natural way to incorporate prior knowledge. In this setting our online

algorithm achieves a sample complexity of $\mathcal{O}(d_{\mathrm{u}}(\ln(N) + \ln(m))/\Delta)$ in terms of policy regret, where $N$ denotes the time horizon, $m$ the number of candidate models, $d_{\mathrm{u}}$ the input dimension, and $\Delta > 0$ a constant that characterizes the separation between models. In the second setting, we allow for any class of dynamical system, where the dynamics are given by a bounded set in a normed vector space. This includes, e.g., all bounded Lipschitz continuous functions with the supremum norm, or a bounded set of square integrable functions. By applying packing and covering arguments, we relate the second setting to the first one and derive corresponding policy-regret guarantees that take the form $\mathcal{O}(N\epsilon^2 + d_{\mathrm{u}}\ln(m(\epsilon))/\epsilon^2)$, where $\epsilon$ describes the discretization width and $m(\epsilon)$ the packing number, which measures the complexity of the function class (Wainwright, 2019). In the third setting, we consider systems that are parametrized by a compact and real-valued set of parameters. This includes the situation where the dynamics are parametrized by neural networks, transformers, or other parametric function approximators, and we obtain a policy regret of $\mathcal{O}(\sqrt{d_{\mathrm{u}}Np})$, where $p$ describes the number of parameters. We further note that in the common situation where our function class is given by a linear combination of nonlinear feature vectors, which also encompasses linear dynamics as a special case, our algorithm is straightforward to implement, as it only requires sampling from a (truncated) Gaussian distribution at every iteration (Cao & Katzfuss, 2024).

Our main contributions are summarized as follows:
- We provide a suite of algorithms with nonasymptotic policy-regret guarantees for online reinforcement learning over continuous state and action spaces with nonlinear dynamics. Numerical results highlight that transients are benign and that our algorithms are likely to be useful in practice.
- Compared to earlier works on posterior sampling reinforcement learning (Osband et al., 2013; Abbasi-Yadkori & Szepesvári, 2015; Xu et al., 2024; Ouyang et al., 2017), our algorithm introduces additional exploration and achieves *frequentist* policy-regret guarantees. Our analysis operates under standard identifiability and persistence of excitation assumptions (Ljung, 1999; Slotine & Li, 1991) suitable for continuous state-action systems even near the stability boundary, which contrasts resampling strategies (Ouyang et al., 2017) or mixing assumptions (Xu et al., 2024).
- Results from earlier works on the linear-quadratic regulator (see, e.g., Dean et al., 2018; Simchowitz & Foster, 2020) are recovered up to constants when specializing to linear dynamics. In contrast to the maximum-a-posteriori identification principle in these works, we rely on posterior sampling, resulting in a Hedge-type algorithm that is general and tractable for analysis.
- Compared to earlier work in the adaptive control community (see, e.g. Anderson et al., 2000; Hespanha et al., 2001), which focuses on asymptotic stability, boundedness, and deterministic systems, we consider stochastic systems and characterize *nonasymptotic performance*. We provide a nonasymptotic bound on the second moment of state trajectories and show that our estimation converges in finite time almost surely.
- The work provides a powerful *separation* principle that applies to nonlinear dynamics. Our algorithms separate optimal model identification and *certainty-equivalent* control, simplifying policy evaluation, e.g., through predictive control or proximal policy optimization with a simulator.

The decision-making problem considered here is central to machine learning and related disciplines, and there has been a great deal of prior work. We provide a short review of recent prior work that is closely related in the following paragraphs; a more detailed literature review can be found in App. A.

The fundamental exploration-exploitation dilemma of reinforcement learning has been elegantly addressed from many angles. Posterior sampling reinforcement learning is an algorithmic paradigm introduced by Osband et al. (2013); Osband & Van Roy (2014); Osband et al. (2016); Osband & Van Roy (2017), which draws analogies to stochastic multi-armed bandits and Thompson sampling (Russo et al., 2018). While these works initially focused on tabular Markov decision processes and episodic settings, extensions to the non-episodic setting have been achieved by Abbasi-Yadkori & Szepesvári (2015); Ouyang et al. (2017); Theocharous et al. (2018); Xu et al. (2024). Our algorithmic idea broadly falls into the category of posterior sampling reinforcement learning, with the distinction of introducing additional excitation and bridging insights from statistics, online learning (Hedge), and control (dissipativity) to provide *frequentist* policy-regret guarantees, whereas these works provide Bayesian guarantees. We further rely on persistence of excitation assumptions, which are common in system identification and statistics (Ljung, 1999; Slotine & Li, 1991). These are weaker than the mixing assumptions of earlier works (see, e.g., Xu et al., 2024).

Optimism in the face of uncertainty is another important principle to balance exploration and exploitation (Lattimore & Szepesvári, 2020). Early works by Auer & Ortner (2006); Bartlett &

Tewari (2009); Jaksch et al. (2010) focus on the tabular setting, while a plethora of recent works (Abbasi-Yadkori & Szepesvári, 2011; Cohen et al., 2019; Abeille & Lazaric, 2020; Croissant et al., 2024; Zanette et al., 2021) demonstrate the usefulness of this principle in handling continuous state-action systems. These approaches maintain confidence sets of parametric models and sample the most optimistic model and policy for exploration. The iterative computation of confidence sets and optimistic policies can be challenging and computationally intensive. Moreover, the resulting complexity measures affecting policy regret, such as the eluder dimension (Foster & Rakhlin, 2023), can be difficult to bound beyond linear Markov decision processes. In contrast, we decouple online best model identification and certainty-equivalent control. This separation not only simplifies policy evaluation (e.g., offline or through model predictive control (Borrelli et al., 2017)), but also facilitates explicit characterization of policy regret via packing numbers, a standard and well-studied complexity measure in statistics.

The research frontier in the domain of reinforcement learning with continuous states and actions (Recht, 2019; Meyn, 2022; Hu et al., 2023) moves from linear (Hazan & Singh, 2022; Tsiamis et al., 2023) to nonlinear dynamics. Prior work (see, e.g., Kakade et al., 2020; Boffi et al., 2021; Lin et al., 2024) shows that sublinear $\mathcal{O}(\sqrt{N})$ regret can be achieved under structural assumptions on the dynamics, e.g., contraction or linear representations of nonlinear features, which are stronger than what is assumed herein. Our approach is also related to multi-model adaptive control, where the goal is asymptotic stabilization (Anderson & Dehghani, 2008), and online switching control (Li et al., 2023; Kim & Lavaei, 2024). In contrast, we provide nonasymptotic policy-regret guarantees that go beyond stabilization and scale with $\mathcal{O}(\ln(m))$ compared to $\mathcal{O}(m^{1/3})$ obtained with switching control. Another important family of methods for learning with continuous states and actions arises from online optimization over an appropriate policy class, such as disturbance-action policies (Agarwal et al., 2019; Hazan et al., 2020; Simchowitz et al., 2020; Li et al., 2021; Chen & Hazan, 2021; Zhao et al., 2023) that are motivated from robust control for linear systems (Gartska & Wets, 1974; Löfberg, 2003; Goulart et al., 2006). Another major approach is to directly optimize over policies, e.g., parametrized via linear functions, by applying zeroth-order or first-order optimization techniques (see, e.g., Fazel et al., 2018; Ma et al., 2024; Hu et al., 2023). Compared to the algorithms and analysis presented herein, the benchmarks in these works are typically different, and focus, for example, on the set of linear disturbance feedback policies or various policy parametrizations that are optimized subject to a fixed system model. The system dynamics are known a priori to the decision-maker, and a tradeoff between exploration and exploitation is absent.

The sample complexity of general interactive decision-making is captured by the Decision-Estimation Coefficient (Foster et al., 2021; 2023; Foster & Rakhlin, 2023), a statistical measure constituting regret lower bounds and supporting an estimation-to-decisions meta algorithm that achieves matching regret upper bounds. This measure generalizes previous frameworks, for instance Bellman rank (Jiang et al., 2017), bilinear classes (Du et al., 2021), and Bellman eluder dimension (Jin et al., 2021), and highlights the interplay among model estimation, model class complexity, and sample efficiency. While the Decision-Estimation Coefficient offers abstract characterizations applicable to general classes of problems and models, we focus on online reinforcement learning problems with continuous states and actions in a non-episodic regime. The challenges herein, such as dependent states, actions, and losses and stability, are not directly handled in the existing Decision-Estimation Coefficient framework. Moreover, nonlinear dynamics may further complicate the specifications of Bellman rank and related variants, which hinge on (linear) function approximation. To address these issues, we leverage tailored tools (e.g., Hedge in online learning and dissipativity in control) and highlight the role of separating best model identification and certainty-equivalent control for sample-efficient reinforcement learning with various model classes (e.g., finite, infinite, and parametric families).

All the aforementioned works provide a basic treatment of online decision-making. However, achieving sublinear policy regret in an online non-episodic setting remains a critical challenge. Our work addresses this challenge in the setting of nonlinear dynamics and smooth non-quadratic stage costs, while recovering earlier $\mathcal{O}(\sqrt{d_{\mathrm{u}}Np})$ regret bounds that are tailored to linear quadratic regulators (e.g., Simchowitz & Foster, 2020), avoiding structural assumptions (Kakade et al., 2020; Boffi et al., 2021; Lin et al., 2024), and greatly improving on the rates from switching control ($\mathcal{O}(\ln(m))$ compared to $\mathcal{O}(m^{1/3})$ or worse). We also differ from previous work on posterior-sampling-based reinforcement learning (Osband et al., 2013) in that we provide *frequentist* policy-regret guarantees and closed-loop stability in an online non-episodic setting. Our approach can further be directly integrated into nonlinear model predictive control techniques (Rawlings et al., 2017; Borrelli et al.,

**Algorithm 1** Reinforcement learning (S1)

**Inputs:** $F := \{f^1, \ldots, f^m\}, \eta, M, \{\sigma_{\mathrm{u}k}^2\}_{k=1}^{\infty}$

   compute $\{\mu^1, \ldots, \mu^m\}$    // e.g. by d. p.
   /* can be approximated by MPC, PPO on simulator*/
  **for** $k = 1, \ldots$ **do**
    // every $M$th step
    **if** $\mathrm{mod}(k-1, M) = 0$ **then**
     $s_k^i \leftarrow \sum_{j=1}^{k-1} \frac{|x_{j+1} - f^i(x_j, u_j)|^2}{1 + |(x_j, u_j)|^2 / b^2}, \forall i \in \{1, \ldots, m\}$
     $i_k \sim \exp(-\eta s_k^i)/Z$
    **else**
     $i_k = i_{k-1}$     //stay with $i_{k-1}$
    **end if**
    //follow policy $i_k$ and add excitation
    $u_k = \mu^{i_k}(x_k) + n_{\mathrm{u}k}, \quad n_{\mathrm{u}k} \sim \mathcal{N}(0, \sigma_{\mathrm{u}k}^2 I)$
  **end for**

**Algorithm 3** Reinforcement learning (S3)

**Inputs:** $F = \{f_\theta \mid \theta \in \Omega\}, \eta, M, \{\sigma_{\mathrm{u}k}^2\}_{k=1}^{\infty}$

  **for** $k = 1, \ldots$ **do**
    // every $M$th step
    **if** $\mathrm{mod}(k-1, M) = 0$ **then**
     $s_k(\theta) = \sum_{j=1}^{k-1} \frac{|x_{j+1} - f_\theta(x_j, u_j)|^2}{1 + |(x_j, u_j)|^2 / b^2}$
     $\theta_k \sim \exp(-\eta s_k(\theta)) \, \mathbb{1}_{\theta \in \Omega}/Z$
     compute $\mu_\theta$ corr. to $f_\theta \in F$   // e.g. by d. p.
     /* can be approximated by MPC, PPO step*/
    **else**
     $\theta_k = \theta_{k-1}$     //stay with $\theta_{k-1}$
    **end if**
    //follow policy $\theta_k$ and add excitation
    $u_k = \mu^{\theta_k}(x_k) + n_{\mathrm{u}k}, \quad n_{\mathrm{u}k} \sim \mathcal{N}(0, \sigma_{\mathrm{u}k}^2 I)$
  **end for**

2017) as also highlighted in App. B and avoids the computation of optimistic policies or confidence regions (Abeille & Lazaric, 2020; Croissant et al., 2024).

The article is structured as follows: Sec. 2 discusses the problem formulation and presents the main results. Sec. 3 illustrates our analysis, where we focus on the first setting (finite set of models)—the other two settings are similar at a high level and we provide a detailed presentation in the appendix. Sec. 4 provides a short conclusion, while proofs, numerical experiments, and further discussion is included in the appendix.

## 2 PROBLEM FORMULATION AND SUMMARY

We consider a reinforcement learning problem where a decision-maker chooses actions $u_k \in \mathbb{R}^{d_{\mathrm{u}}}$ to control a dynamical system $x_{k+1} = f(x_k, u_k) + n_k$, where $x_k \in \mathbb{R}^{d_{\mathrm{x}}}$ denotes the state, $f : \mathbb{R}^{d_{\mathrm{x}}} \times \mathbb{R}^{d_{\mathrm{u}}} \to \mathbb{R}^{d_{\mathrm{x}}}$ the dynamics (unknown to the decision-maker), and $n_k \sim \mathcal{N}(0, \sigma^2 I)$ the process noise (independent across time; can be non-zero-mean or sub-Gaussian, see below). Without loss of generality, we set $x_1 = 0$. We further denote the Lipschitz constant of $f$ in $(x, u)$ by $L$.

The decision-maker aims at minimizing the expected loss, $\mathrm{E}[\sum_{k=1}^{N} l(x_k, u_k)]$, where $l : \mathbb{R}^{d_{\mathrm{x}}} \times \mathbb{R}^{d_{\mathrm{u}}} \to \mathbb{R}_{\geq 0}$ captures the stage cost, by learning and applying an appropriate and possibly random feedback policy $u_k = \mu_k(x_k)$.

We consider three different settings. In the first setting (S1) the decision-maker has access to $m$ (nonlinear) candidate models $F := \{f^1, \ldots, f^m\}, f^i : \mathbb{R}^{d_{\mathrm{x}}} \times \mathbb{R}^{d_{\mathrm{u}}} \to \mathbb{R}^{d_{\mathrm{x}}}, i = 1, \ldots, m$ that describe potential system dynamics. Each $f^i$ is $L$-Lipschitz in $(x, u)$. In the second setting (S2), we allow for any class of functions $F$ that is given by a bounded set in a normed vector space, which is therefore much broader and includes, for example, all bounded $L$-Lipschitz functions with the usual supremum norm. In the third setting (S3), the dynamics are parametrized by the parameter $\theta$, i.e., $F = \{f_\theta(x, u) \mid \theta \in \Omega\}$, where $\Omega$ is a compact real-valued set. Without loss of generality, we assume that $\Omega$ is contained in a unit ball by scaling the parameters accordingly. This captures the setting where the functions $f_\theta$ are represented by neural or transformer architectures, or when $f_\theta$ are given by linear combinations of (nonlinear) feature vectors $f_\theta(x, u) = \theta^\top \phi(x, u)$. This also encompasses linear dynamics as a special case. We further assume that the system dynamics $f$ are contained in the set of candidate models, i.e., $f \in F$ for each setting. This realizability assumption can be easily relaxed as discussed in App. G, provided that there is a single candidate model that is closest to $f$ on the entire state-action space.

This article analyzes the decision-making strategy listed in Alg. 1, which can be easily adapted to the settings S2/S3 (see Alg. 2 in App. D and Alg. 3). The algorithm keeps track of the one-step prediction error (Ljung, 1999; Chua et al., 2018; Janner et al., 2019), that is,

$$s_k^i = \sum_{j=1}^{k-1} \frac{|x_{j+1} - f^i(x_j, u_j)|^2}{1 + |(x_j, u_j)|^2 / b^2}, \quad f^i \in F, \tag{1}$$

where $b > 0$ is a sufficiently large constant, and $|(x_j, u_j)|$ denotes the $\ell_2$-norm of a vector stacking $x_j$ and $u_j$. The normalization with $1 + |(x_k, u_k)|^2/b^2$ ensures that the variables $s_k^i$ remain bounded even when $x_k, u_k$ become arbitrarily large, while for small $x_k, u_k$ the normalization is close to the identity. This will simplify the subsequent analysis and the resulting statement of the policy-regret bounds. Our analysis also carries over to the limiting case where $b$ is unbounded (i.e., $b \to \infty$), and the same regret bounds apply, as is discussed in App. F; however, the constants in the resulting regret guarantees and algorithm parameters become more complex. The sum of squared distances $|x_{j+1} - f^i(x_j, u_j)|^2$ can be interpreted as the negative log-likelihood of model $i$ given the past trajectory $\{x_j, u_j\}_{j=1}^k$, due to the Gaussian process noise. Hence, from a Bayesian perspective, the distribution $\exp(-s_k^i)$ represents the probability that model $f^i$ corresponds to $f$ given the past trajectory. The scaling with $\eta$ implements a softmax (for $\eta$ large we greedily pick the model that maximizes the posterior, for $\eta \approx 1$ we directly sample from the posterior). The update rule has close connections to Hedge or multiplicative weights, which is a common decision-making strategy in online learning (Cesa-Bianchi & Lugosi, 2006; Arora et al., 2012; Mourtada & Gaïffas, 2019). Nonetheless, our setting is mathematically distinct from Hedge due to dynamics that couple states, actions, and rewards across time, and the need for exploration. As such, existing analysis techniques do not apply. Furthermore, our analysis extends to non-zero-mean $n_k$, since this can be captured by modifying $f$ accordingly, and generalizes to sub-Gaussian process noises thanks to the corresponding bounds on moment-generating functions.

The algorithm chooses control actions $u_k$ as

$$u_k = \mu^{i_k}(x_k) + n_{\mathrm{u}k},$$

where $n_{\mathrm{u}k} \sim \mathcal{N}(0, \sigma_{\mathrm{u}k}^2 I)$, and $i_k$ is a random variable that is defined in the following way: If $\mathrm{mod}(k-1, M) = 0$, $i_k$ takes the value $i_k = i$ with probability density $p_k^i \sim \exp(-\eta s_k^i)/Z$ (conditional on the past), where $Z$ denotes a normalization constant. If $\mathrm{mod}(k-1, M) \neq 0$, $i_k$ remains fixed, i.e., $i_k = i_{k-1}$. The random variable switches only every $M$th step, which ensures that the excitation with $n_{\mathrm{u}k}$ is rich enough, as specified precisely in Ass. 3 below. The feedback policy $\mu^i$ describes any policy associated with candidate model $f^i$, i.e., a policy that achieves the performance

$$\limsup_{N \to \infty} \frac{1}{N} \mathrm{E}\Big[\sum_{k=1}^{N} l(x_k^i, \mu^i(x_k^i))\Big] = \gamma^i, \tag{2}$$

on the candidate model $f^i$, where $x_{k+1}^i := f^i(x_k^i, \mu^i(x_k^i)) + n_k$ with $x_1^i = 0$. The policy $\mu^i$ can be optimal for model $f^i$, but this does not necessarily need to be the case. In practice, such a policy can be obtained by solving a Bellman equation through (approximate) dynamic programming (Bertsekas, 2017), or by applying proximal policy optimization in conjunction with an offline simulator. We assume $\gamma^i$ (i.e., the steady-state performance of $\mu^i$ associated with $f^i$) to be finite for all $i$. If $\gamma^i$ is infinite, then the corresponding model $f^i$ should be excluded. We will consider policy regret as our performance objective, where the policy $\mu$ corresponding to the dynamics $f$ represents the benchmark performance.

The reinforcement learning strategy has a natural interpretation: The strategy selects, at each $M$th iteration, the feedback policy $\mu^{i_k}$, where the index $i_k$ is sampled from a distribution following a softmax function of $s_k^i$. The system is further excited by adding the random perturbation $n_{\mathrm{u}k}$ to the feedback policy. If persistence of excitation is guaranteed, the estimation will converge at a rate at least $\mathcal{O}(1/k^2)$, which yields a policy regret (compared to the strategy $\mu$ corresponding to the dynamics $f$) that scales logarithmically in the horizon $N$ and the number of candidates $m$.

We emphasize that our analysis technique translates in straightforward ways to more general situations than the ones described herein. For example, while this article focuses on time-invariant policies, it would be straightforward to also incorporate time-varying policies $\mu_k^i$, and a corresponding finite-horizon benchmark. More precisely, we focus on steady-state performance, where the benchmark is given by the steady-state performance of policy $\mu$ (corresponding to $f$). However, finite-horizon objectives can be easily accommodated by measuring regret with respect to the optimal finite-horizon policy $\mu_k$; the same nonasymptotic regret bounds would apply. The article also focuses on "naive" excitation signals $n_{\mathrm{u}k}$, sampled from a normal distribution. However, our analysis principle is flexible enough to also incorporate more general type of excitation strategies (e.g., relying on domain-specific knowledge), as long as the excitation has finite second moments and guarantees a persistence condition similar to Ass. 3.

Our results are summarized as follows:

**Theorem 2.1** *(S1) Let the cost-to-go function corresponding to $f$ and the stage cost $l$ be smooth (see Ass. 1 and 2), the feedback policies $\mu^i$ be Lipschitz continuous, and let a persistence of excitation condition be satisfied (see Ass. 3). Then, for a constant learning rate $\eta$ and $\sigma_{uk}^2 \sim 1/(\Delta k) + \ln(m)/(\Delta k^2)$ the policy regret of Alg. 1 is bounded by*

$$\mathrm{E}[\sum_{k=1}^{N} l(x_k, u_k)] - N\gamma \le c_{r1}d_u\ln(N)/\Delta + c_{r2}d_u\ln(m)/\Delta + c_{r3}\sigma^2 d_x + c_{r1}d_u/\Delta,$$

*for all $N \ge 2M$, where $c_{r1}, c_{r2}, c_{r3}$ are constant, $\gamma$ corresponds to the steady-state performance related to $f$ (see (2)), and $\Delta$ characterizes the discrepancy between models. The precise constants are listed in Thm. 3.2.*

**Theorem 2.2** *(S2) Let the set of candidate models $F$ be a bounded set in a normed vector space. Let the cost-to-go function corresponding to $f$ and the stage cost $l$ be smooth (see Ass. 2 and 5), the feedback policies $\mu^{\bar{f}}$ corresponding to an $\bar{f} \in F$ be Lipschitz continuous, and let a persistence of excitation condition be satisfied (see Ass. 4). Then, for all $N \ge 2M$, any $\epsilon > 0$, for a constant learning rate $\eta$, and $\sigma_{uk}^2 \sim 1/(\epsilon^2 k) + \ln(m(\epsilon))/(\epsilon^2 k^2)$, the policy regret of Alg. 2 (see App. D) is bounded by*

$$\mathrm{E}[\sum_{k=1}^{N} l(x_k, u_k)] - N\gamma \le c_{r0}N\epsilon^2 + c_{r1}d_u\ln(N)/\epsilon^2 + c_{r2}d_u\ln(m(\epsilon))/\epsilon^2 + c_{r3}\sigma^2 d_x,$$

*where $m(\epsilon)$ denotes the packing number of the set $F$. The precise constants are listed in Thm. D.2.*

**Theorem 2.3** *(S3) Let the set of candidate models $F$ be parametrized by $\theta$, i.e., $F = \{f_\theta(x, u) \mid \theta \in \Omega\}$, where $\Omega \subset \mathbb{R}^p$ is contained in a unit ball of dimension $p$. Let the cost-to-go function corresponding to $f$ and the stage cost $l$ be smooth (see Ass. 2 and Ass. 7), the feedback policies $\mu_\theta$ corresponding to each $f_\theta \in F$ be Lipschitz continuous, and let a persistence of excitation condition be satisfied (see Ass. 6). Then, for all $N \ge 2M$, for a constant learning rate $\eta$, and $\sigma_{uk}^2 \sim \sqrt{p/(kd_u)}$, the policy regret of Alg. 3 is bounded by*

$$\sum_{k=1}^{N} \mathrm{E}[l(x_k, u_k)] - N\gamma \le \sqrt{(c_{r1}\ln(N) + c_{r2}p)d_u N} + c_{r3}\sigma^2 d_x.$$

*The precise constants are listed in Thm. E.1.*

The above theorems follow from Thms. 3.2, D.2, and E.1 by renaming constants. We refer to Sec. 3.1 and the appendix for formal proofs. The results characterize precisely how the policy regret scales with the dimension $d_x, d_u$ and the time horizon $N$. In the setting of Thm. 2.1, we have a finite class of models, and the policy regret scales with $\ln(m)$, which is in line with the literature on online learning (Cesa-Bianchi & Lugosi, 2006; Lattimore & Szepesvári, 2020).[1] Thm. 2.2 relies on a packing argument, whereby the set $F$ is successively approximated by a finite number of candidate models. The result is stated in full generality; for a specific function class $F$ and packing number $m(\epsilon)$ the right-hand side can be minimized over $\epsilon$ (the discretization width). For instance if $F$ consists of the space of bounded $L$-Lipschitz functions, the packing number $m(\epsilon)$ scales with $d_x \exp((L/\epsilon)^{d_x+d_u})$, which means that the policy regret grows roughly with $N^{(d_x+d_u)/(d_x+d_u+2)} = o(N)$, and establishes no-regret learning for a very large class of functions. In the special case where $d_x = d_u = 1$, the right-hand side grows with $\sqrt{N}$. Thm. 2.3 is of direct practical importance, since it provides an algorithm and corresponding regret bound that applies to the typical scenario where the functions $F$ are parametrized, for example by neural networks. In the simplest setting, $F$ consists of linear dynamical systems, which directly recovers well-known results from the literature (e.g., Dean et al., 2018; Mania et al., 2019; Simchowitz & Foster, 2020). More precisely, if $F$ consists of linear

---

[1]The bound depends on $\Delta$, characterizing the discrepancy between models. If models are arbitrarily close to each other, the bound applies asymptotically and becomes loose for small $m$ and $N$. In this situation, a tighter bound of $\mathcal{O}(\sqrt{d_u N \ln(m)})$ is achieved for small $m$, $N$, by distinguishing models to ensure $\Delta \sim \sqrt{d_u \ln(m)}/\sqrt{N}$. The situation is captured by Thm. 2.2 and Thm. 2.3 and is therefore not discussed further.

dynamical systems, the number of parameters is given by $d_x^2 + d_x d_u$, which means that the resulting regret bound scales with $\sqrt{(d_x^2 + d_x d_u)d_u N}$.

Our algorithms are optimal up to logarithmic factors, since the formulation incorporates online regression as a special case. This results in an $\Omega(d_u \ln(m))$ lower bound for setting (S1) and a $\Omega(\sqrt{N d_u p})$ lower bound for setting (S3) (Rakhlin & Sridharan, 2014).

We conclude the summary by commenting on boundedness of states. In control-theoretic applications (and in the related community) boundedness of solutions and benign transients are a primary concern. We will see that in all our results we can ensure boundedness provided that the stage cost satisfies $l(x, u) \geq \underline{L}_l |x|^2/2$ for a constant $\underline{L}_l > 0$. More precisely, we can guarantee that

$$\underline{L}_l \mathrm{E}[|x_k|^2] \leq 2\mathrm{E}[V(x_k)] \leq c_b \tag{3}$$

for all $k = 1, \ldots$, along the trajectories of our reinforcement learning algorithm, where $V$ refers to the cost-to-go function corresponding to the dynamics $f$ and policy $\mu$, and $c_b > 0$ is an explicit constant. Due to the fact that the dynamics are Lipschitz continuous and $n_k, n_{uk}$ are Gaussian, $x_k, u_k$ are in fact sub-Gaussian with mean and second moment bounded by $\sqrt{c_b/\underline{L}_l}$ and $c_b/\underline{L}_l$, respectively, and we can therefore characterize tail probabilities for finite $k$, as well as for arbitrarily large values of $k$ under ergodicity assumptions.

## 3 SUMMARY OF THE ANALYSIS

This section discusses the technical details and insights that lead to the results presented in Thm. 2.1-2.3. The presentation focuses on setting S1, since the results in setting S2 and S3 follow analogously.

### 3.1 FINITE MODEL SET-UP

This section considers $F = \{f^1, \ldots, f^m\}$ being finite and $f \in F$. We denote the cost-to-go function related to the dynamics $f$ and the policy $\mu$ by $V : \mathbb{R}^{d_x} \to \mathbb{R}_{\geq 0}$, where $V$ is any function that satisfies the following assumption:

**Assumption 1** *(Bellman-type inequality) The cost-to-go function $V$ (corresponding to $f$ and $\mu$) satisfies the following inequality*

$$V(x) \geq \mathrm{E}[l(x, u) + V(f(x, u) + n)] - \gamma - d_u L_u \sigma_u^2, \tag{4}$$

*for a constant $L_u$ and for all $x \in \mathbb{R}^{d_x}$, where $u = \mu(x) + n_u$, $n \sim \mathcal{N}(0, \sigma^2 I)$, $n_u \sim \mathcal{N}(0, \sigma_u^2 I)$, and the expectation is taken over $n$ and $n_u$.*

Ass. 1 is met for linear dynamical systems (Abeille & Lazaric, 2020; Abbasi-Yadkori & Szepesvári, 2011; Simchowitz & Foster, 2020; Dean et al., 2018) and nonlinear dynamical systems under dissipation assumptions (Khalil, 2002, Ch. 5), e.g., Boffi et al. (2021); Li et al. (2023). The rationale behind Ass. 1 is the following: From a dynamic programming point of view computing an optimal policy $\mu$ requires solving a corresponding infinite-horizon average-cost-per-stage problem. In general, a corresponding Bellman equation and cost-to-go function might not exist, as for example discussed in Bertsekas (2017), Ch. 5. The formulation via Ass. 1 circumvents these technical difficulties, due to the fact that $\gamma$ is not required to correspond to the optimal infinite-horizon average cost. Indeed, from a control-theoretic point of view Ass. 1 characterizes a notion of dissipation (Willems, 2007), where $V$ represents a storage function and $-l(x, u) + \gamma$ the supply rate (for $\sigma_u = 0$). Moreover, if a Bellman equation (Bertsekas, 2017, Prop. 5.5.1) and corresponding cost-to-go function exist for the dynamics $f$, then Ass. 1 is clearly satisfied for the corresponding cost-to-go function (for $\sigma_u = 0$). The additional term $d_u L_u \sigma_u^2$ captures the influence of the excitation $n_u$ and is without loss of generality, since for any smooth function $\xi : \mathbb{R}^{d_u} \to \mathbb{R}$ the following applies

$$\mathrm{E}[\xi(u + n_u)] = \mathrm{E}\Big[\xi(u) + \nabla \xi(u)^\top n_u + \frac{1}{2} n_u^\top \nabla^2 \xi(\bar{u}) \, n_u\Big] = \mathrm{E}[\xi(u)] + \mathcal{O}(d_u \sigma_u^2),$$

where $n_u \sim \mathcal{N}(0, \sigma_u^2 I)$. The constant $L_u$ in Ass. 1 makes the previous bound quantitative. Ass. 1 excludes dynamics where a slight perturbation of the input away from the policy $\mu$ leads to an unbounded cost.[2]

---

[2] A simple example is $x_k = \exp(\exp(u_k^2))$, $\mu = 0$, $l(x, u) = x^2$ where $\mathbb{E}[\exp(\exp(n_u^2))^2]$ is unbounded.

We will further require the following smoothness conditions:

**Assumption 2** *The policies $\mu^i$ are $L_\mu$ Lipschitz, the stage-cost $l$ is $\bar{L}_l$ smooth, and the cost-to-go function $V$ is $\bar{L}_V$ smooth and satisfies $V(x) \geq -\underline{c}_V + \underline{L}_V |x|^2/2$ for some $\underline{L}_V > 0$ and $\underline{c}_V \geq 0$.*

Ass. 2 is met for linear dynamical systems (Abeille & Lazaric, 2020; Abbasi-Yadkori & Szepesvári, 2011; Simchowitz & Foster, 2020; Dean et al., 2018) and many nonlinear dynamical systems, e.g., (Khalil, 2002, Ch. 5) when $l$ is a positive definite quadratic. These smoothness assumptions will be needed to analyze how the cost-to-go $V$ evolves if the feedback policy $\mu^q \neq \mu$ is applied and prevent the state from diverging in finite time. We note that the quadratic lower bound on $V(x)$ is automatically satisfied in view of Ass. 1 if $l(x, u) \geq \underline{L}_l |x|^2/2$ for a constant $\underline{L}_l > 0$.

We further require the following assumption:

**Assumption 3** *There exists an integer $M > 0$ and two constants $\Delta > 0$ and $b > 0$ such that for any $x_1 \in \mathbb{R}^{d_x}$, $\sigma_u > 0$, and $f^i \in F$, $f^i \neq f$,*

$$\frac{1}{M} \sum_{k=1}^{M} \mathrm{E}\Big[\frac{|f^i(x_k, u_k) - f(x_k, u_k)|^2}{1 + |(x_k, u_k)|^2/b^2}\Big] \geq \Delta \sigma_u^2$$

*holds, where $x_{k+1} = f(x_k, u_k) + n_k$, $u_k = \mu^q(x_k) + n_{uk}$ with $n_k \sim \mathcal{N}(0, \sigma^2 I)$, $n_{uk} \sim \mathcal{N}(0, \sigma_u^2 I)$, and $q \in \{1, \ldots, m\}$.*

The previous assumption specifies persistence of excitation, which guarantees that the estimate $i_k$ of the best candidate model will quickly converge to $i^*$, where $f^{i^*} = f$, and is standard in the statistics (Fisher information) and system identification literature (see, e.g., Ljung, 1999, Ch. 8.2); (Slotine & Li, 1991; Ly et al., 2017; Chatzikiriakos et al., 2025). Ass. 3 is generically satisfied for linear systems with $M = 2$, whereby the constant $\Delta$ relates to the controllability of the closed-loop dynamics and the accuracy $|A^i - A|_F^2$ and $|B^i - B|_F^2$ of the different candidate models with $|\cdot|_F$ the Frobenius norm, see App. F. The assumption is also met for a broad class of nonlinear dynamical systems, as shown in App. F and Prop. F.2. Ass. 3 includes a normalization with the constant $b$, whereas the literature usually considers $b \to \infty$. However, as discussed in App. F our analysis also encompasses the case $b \to \infty$; the resulting constants are more elaborate and we therefore focus our discussion on the situation where $b$ is finite. Moreover, Ass. 3 is a more general version of the "uniformly excited feature" assumption, which is common in online reinforcement learning (Hao et al., 2021; Liu et al., 2023; Lazic et al., 2020) and arises from $f^i(x, u) = \phi(x, u)^\top \theta^i$ and setting $M = 1$.

Our analysis of Alg. 1 starts by showing that the convergence to the best candidate model is fast, which leads to the logarithmic scaling of the policy regret with $N$ and $m$. This is summarized with the following proposition:

**Proposition 3.1** *Let Ass. 3 be satisfied and let the step size be $\eta \leq \min\{1/(4M\sigma^2), 1/(2ML^2 b^2)\}$. Then, the following holds*

$$\Pr(i_k = i) \leq \exp\left(-\frac{\Delta \eta}{4} \sum_{j=1}^{k-M} \sigma_{uj}^2\right),$$

*for $k = 1, 2, \ldots$ and any $i \neq i^*$, where $f^{i^*} = f$. Moreover, it holds that*

$$\Pr(i_k \neq i^*) \leq \frac{M^2}{(k-M)^2}, \qquad \forall k \geq M+1$$

*for $\sigma_{uk}^2 = \frac{4}{\eta \Delta M}\left(\frac{2}{\lceil k/M \rceil} + \frac{\ln(m)}{(\lceil k/M \rceil)^2}\right)$, where $\lceil \cdot \rceil$ denotes rounding to the next higher integer.*

**Proof** The proof can be found in App. C.1 and relies on a concentration of measure argument. $\square$

An immediate corollary of the fast convergence rate established with Prop. 3.1 is that the sequence $i_k$ will converge to $i^*$ in finite time (almost surely), where $f^{i^*} = f$. This is discussed in Cor. C.6. As a result of Prop. 3.1, we are now ready to state and prove our first main result that characterizes the policy regret in setting S1.

**Theorem 3.2** *Let Ass. 1, 2 and 3 be satisfied and choose $\eta \leq \min\{1/(4M\sigma^2), 1/(2ML^2b^2)\}$ and $\sigma_{uk}^2$ as in Prop. 3.1. Then, the policy regret of Alg. 1 is bounded by*

$$\sum_{k=1}^{N} \mathrm{E}[l(x_k, u_k)] - N\gamma \leq c_{r1} + c_{r2}Md_u\ln(m)/\Delta + c_{r2}d_u\ln(N)/\Delta,$$

*for all $N \geq 2M$, where the constants $c_o, c_2$ are specified in Lemma C.3, and $c_{r1}, c_{r2}$ are given by*

$$c_{r1} = 3c_\alpha M(d_x \sigma^2 \bar{L}_V/2 + c_o) + c_{r2}d_u/\Delta, \quad c_{r2} = 8c_\alpha(\bar{L}_V L^2 + \bar{L}_l + L_u)/\eta, \quad c_\alpha = e^{3c_2 M}.$$

**Proof** (Sketch, details are in App. C.2) The proof relies on using $V$ as a Lyapunov function and performing the following decomposition

$$\mathrm{E}[V(x_{k+1})] = \mathrm{E}[V(x_{k+1})|i_k \neq i^*]\mathrm{Pr}(i_k \neq i^*) + \mathrm{E}[V(x_{k+1})|i_k = i^*]\mathrm{Pr}(i_k = i^*). \quad (5)$$

The first term describes the evolution of $V(x_{k+1})$ when choosing $i_k \neq i$, and in this (unfavorable) situation $V$ may grow at most exponentially. This is captured by the following bound that relies on the continuity assumptions on $V$ (see Lemma C.3)

$$\mathrm{E}[V(x_{k+1})|i_k \neq i^*] \leq c_2\mathrm{E}[V(x_k)] + \mathcal{O}(\sigma^2 + \sigma_{uk}^2) - \mathrm{E}[l(x, u_k)|i_k \neq i^*],$$

where the notation $\mathcal{O}$ hides continuity and dimension-related constants. The second term in (5), describes the favorable situation of choosing $i_k = i^*$, where $V(x_{k+1})$ is bounded as a result of the Bellman-type inequality (4). This yields:

$$\mathrm{E}[V(x_{k+1})|i_k = i^*] \leq \mathrm{E}[V(x_k)] + \gamma + \mathcal{O}(\sigma_{uk}^2) - \mathrm{E}[l(x_k, u_k)|i_k = i^*],$$

where continuity and dimension-related constants are again hidden. By combining the two inequalities we arrive at

$$\mathrm{E}[V(x_{k+1})] \leq \mathrm{E}[V(x_k)](c_2\mathrm{Pr}(i_k \neq i^*) + 1) + \gamma - \mathrm{E}[l(x_k, u_k)] + \mathcal{O}(\sigma_{uk}^2 + \mathrm{Pr}(i_k \neq i^*)\sigma^2). \quad (6)$$

From Prop. 3.1, we know that $\mathrm{Pr}(i_k \neq i^*)$ decays at rate $1/k^2$. This means that, roughly speaking, the inequality (6) gives rise to a telescoping sum (see Lemma C.4 for details), which yields

$$\sum_{k=1}^{N} \mathrm{E}[l(x_k, u_k)] - \gamma N \leq \mathcal{O}(\sum_{k=1}^{N}(\sigma_{uk}^2 + \mathrm{Pr}(i_k \neq i^*)\sigma^2)).$$

The fact that $\mathrm{Pr}(i_k \neq i^*)$ is summable, due to the decay at rate $1/k^2$, and that the sum over $\sigma_{uk}^2$ evaluates to $\mathcal{O}(\ln(N) + \ln(m))$ establishes the desired result up to constants (these are computed in App. C.2). □

The proof of Thm. 3.2 relies on using $V$ as a Lyapunov function. Provided that the stage cost $l(x, u)$ is bounded below by a quadratic of the type $|x|^2$, we can modify the analysis in straightforward ways to obtain explicit bounds on $\mathrm{E}[V(x_k)]$ and hence on $\mathrm{E}[|x_k|^2]$, uniform over $k$, which is an important concern in the adaptive control community. Moreover, these bounds require persistence of excitation only over a finite number of steps, since $\mathrm{Pr}(i_k \neq i^*)$ is monotonically decreasing even when Ass. 3 is not satisfied. The details are presented in App. C.4.

### 3.2 INFINITE CARDINALITY

The ideas described in the previous section translate to the situation in which the set of candidate models $F$ is a bounded subset of a normed vector space with norm $\|\cdot\|$. For example, $F$ could represent the set of bounded, $L$-Lipschitz continuous functions that map from $\mathbb{R}^{d_x} \times \mathbb{R}^{d_u} \to \mathbb{R}^{d_x}$, with $\|\cdot\|$ the supremum norm. Alternatively, $F$ could be a bounded subset of the set of square integrable functions. Our purpose is to provide an upper bound on the *sample complexity* of online reinforcement learning in this very general setting and not to characterize *computational complexity*; see the next subsection for a computationally tractable variant. As such, the results herein center on learnability guarantees rather than direct deployment of online reinforcement learning for a continuum of models.

The key insight is that S2 can be reduced to S1 by constructing an $\epsilon$-packing $F_k^\epsilon$ of the set $F$ in an online way, by greedily adding functions $f^i \in F$ as long as $\|f^i - \bar{f}\| > \epsilon$ for all $\bar{f} \in F_k^\epsilon$.

As a result $F_k^\epsilon$ covers $F$ by construction, i.e., for every $f \in F$ there exists $f^i \in F_k^\epsilon$ such that $\|f^i - f\| \leq \epsilon$. The cardinality of $F_k^\epsilon$ is bounded by the packing number of $F$, which is denoted by $m(\epsilon)$. The construction needs to be done online in order to include the minimizer $\mathrm{argmin}_{\bar{f} \in F} s_k(\bar{f})$ in $F_k^\epsilon$, where $s_k(\bar{f})$ denotes the prediction error as before, accumulated over $k - 1$ steps (see (1)). This means that the arguments used in deriving Prop. 3.1 apply in the same way and implies that $\Pr(i_k \notin I_k^*) \leq M^2/(k - M)^2$ as before, where $I_k^*$ denotes the set of models $f^{i*} \in F_k^\epsilon$ that satisfy $\|f^{i^*} - f\| \leq \epsilon$. As a result, the same arguments as in the proof of Thm. 3.2 apply, which yields the statement of Thm. 2.2. The details of the corresponding algorithm (i.e., Alg. 2) and analysis are presented in App. D.

### 3.3 PARAMETRIC MODELS

The following section discusses the situation where the set of candidate models $F$ is parametrized by a parameter $\theta \in \Omega \subset \mathbb{R}^p$, where $\Omega$ is contained in a $p$-dimensional unit ball, that is,

$$F = \{f_\theta : \mathbb{R}^{d_x} \times \mathbb{R}^{d_u} \to \mathbb{R}^{d_x} \mid \theta \in \Omega\}.$$

The canonical example we have in mind is when $f_\theta$ is parametrized with a large neural network, transformer, or state-space architecture, where $\theta$ represents the parameters. As in the previous section, we assume that $f \in F$, and without loss of generality, we set $f = f_{\theta=0}$, i.e., the parameters are centered around $f$.

Alg. 3 has a particularly straightforward interpretation, which also facilitates its implementation in practice. In each iteration, $f_{\theta_k}$ is sampled from the posterior distribution over models $f_\theta$, scaled by $\eta$. In the special case where $f_\theta(x, u) = \phi(x, u)^\top \theta$ we note that the density $\exp(-\eta s_k(\theta))/Z$ corresponding to the random variable $\theta_k$ is Gaussian, with mean and covariance

$$\underset{\theta \in \mathbb{R}^p}{\mathrm{argmin}} \sum_{j=1}^{k-1} \frac{|x_{j+1} - \phi(x_j, u_j)^\top \theta|^2}{1 + |(x_j, u_j)|^2/b^2}, \quad \frac{1}{2\eta} \left( \sum_{j=1}^{k-1} \frac{\phi(x_j, u_j)\phi(x_j, u_j)^\top}{1 + |(x_j, u_j)|^2/b^2} \right)^{-1}.$$

The Gaussian mean and covariance can be efficiently evaluated by running a recursive least squares algorithm, resulting in a per-iteration computational complexity of only $\mathcal{O}(p^2)$. The corresponding computation of the policy $\mu_\theta$ for the model $f_\theta$ is more challenging, but can, in principle, be done offline with dynamic programming, or in an offline simulation with proximal policy optimization, for example. A notable exception is when $\phi(x, u)$ is linear, in which case the corresponding (steady-state optimal) policy $\mu_\theta$ is linear and can be computed by solving a Riccati equation in $\mathcal{O}(d_x^3)$ steps. If $f_\theta$ has a more general structure, the sampling can, for example, be implemented with Langevin Monte-Carlo (Vempala & Wibisono, 2019; Jeon et al., 2025). The regret analysis follows the same steps as in Sec. 3.1 and is included in App. E.

Numerical examples that illustrate Alg. 1 and Alg. 3 on linear and nonlinear systems are included in App. B and highlight a rapid convergence of the posterior distribution over models, as well as, the implementation of the policy evaluation via model predictive control. All experiments run in a few minutes on a laptop. For nonlinear systems, the bottleneck of Alg. 1 is found to be in the policy evaluation, which does not depend on the number of candidate models, rather than the posterior sampling over models (even when scaling to 10,000 models).

## 4 CONCLUSION

This article provides policy-regret guarantees for online reinforcement learning with *nonlinear dynamical systems over continuous state and action spaces*. We provide a suite of algorithms and prove that the resulting policy regret over $N$ steps scales as $\mathcal{O}(d_u \ln(N)/\Delta + d_u \ln(m)/\Delta)$ in a setting where there is a finite class of $m$ models that are separated via the constant $\Delta$ and as $\mathcal{O}(\sqrt{d_u N p})$ in a setting where models are parametrized over a compact real-valued space of dimension $p$. The results require persistence of excitation, and rely on continuity assumptions on the dynamics.

The results highlight important and fruitful connections between reinforcement learning and control theory and open numerous exciting future research avenues. Technical extensions include i) developing a computationally tractable algorithm for setting (S2) through hierarchical coverage of model classes, ii) designing exploration strategies beyond additive Gaussian noise that guarantee persistence of excitation and iii) handling partial observability and non-additive noise in dynamics.

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

## A  RELATED WORK

This article revolves around online reinforcement learning with multiple nonlinear candidate models. We adopt a viewpoint at the intersection of online decision-making, reinforcement learning, and adaptive control. We review representative works along major distinct algorithmic ideas as follows.

**Posterior sampling reinforcement learning** has been introduced by Osband et al. (2013); Osband & Van Roy (2014); Osband et al. (2016); Osband & Van Roy (2017), where sample complexity of online reinforcement learning is typically quantified via the eluder dimension. These works focus on episodic reinforcement learning and draw analogies to stochastic multi-armed bandits and Thompson sampling (Russo et al., 2018). The relation to Thompson sampling enables algorithms that are effective in practice and, compared to optimistic approaches discussed below, avoid the computation of confidence regions, which can be nontrivial. These initial works focus on episodic and finite-state-action settings. The resulting policy-regret guarantees are Bayesian (so-called Bayesian regret), i.e., the regret bounds do not hold for any potential environment in the set of candidates, but only in expectation, when environments are sampled according to the prior. Our work establishes frequentist policy-regret guarantees by introducing additional exploration. One line of follow-up works by Abbasi-Yadkori & Szepesvári (2015); Abeille & Lazaric (2017); Kargin et al. (2022) deals with continuous states and actions by exploiting linear parameterizations of dynamics and/or controllers. Another line of works by Fan & Ming (2021); Sasso et al. (2023) leverages approximations of dynamics, rewards, or value functions to navigate through continuous state-action spaces. Model-free variants of posterior-sampling reinforcement learning (Osband et al., 2016; 2019; Janz et al., 2019) maintain a distribution of value functions (instead of dynamics), albeit entailing challenges in explicit representations and accurate updates of such value functions. The online non-episodic scenario, most relevant to our work, is addressed in Abbasi-Yadkori & Szepesvári (2015); Theocharous et al. (2018); Xu et al. (2024). Along this line, Xu et al. (2024) augment existing posterior sampling reinforcement learning with an environment resampling at random Bernoulli trials. This addition generalizes the theory to high-dimensional weakly communicating Markov decision processes and continuous state-action spaces via function approximation in non-episodic settings.

Our algorithmic paradigm falls into the category of posterior sampling reinforcement learning. Different from existing works, we bridge insights from statistics (Hedge updates) and control (dissipativity and Lyapunov analysis) to handle dependent states, actions, and losses in online non-episodic reinforcement learning with continuous states and actions. This bridge enables us to explicitly establish *frequentist* policy regret guarantees, closed-loop stability, and bounded losses in the presence of complex coupling arising from non-stationary and nonlinear dynamics. Our policy-regret guarantees are subject to a persistence of excitation/identifiability assumption as introduced in the system identification and statistics literature (Ljung, 1999, Ch. 8.2); (Slotine & Li, 1991), which are weaker than the mixing assumptions from Xu et al. (2024) that, e.g., do not apply to dynamics close to the stability boundary.

**Optimism in the face of uncertainty** is a well-known paradigm from the multi-armed bandit literature that inherently trades off exploration with exploitation. The paradigm has been applied to online reinforcement learning in the tabular setting by Auer & Ortner (2006); Bartlett & Tewari (2009); Jaksch et al. (2010), and later extended to more general Markov decision processes by Abbasi-Yadkori & Szepesvári (2011); Cohen et al. (2019); Abeille & Lazaric (2020); Croissant et al. (2024); Kakade et al. (2020); Zanette et al. (2021); Yang & Wang (2020); Jin et al. (2023). Later approaches hinge on iteratively refining parametric models with confidence bounds and applying policies associated with the most optimistic model. The resulting frequentist regret bounds typically enjoy a square-root dependence on the time horizon (except for Zanette et al. (2021)) and introduce different complexity measures that capture the function class at hand. The work by Yang & Wang (2020); Jin et al. (2023) focuses on an episodic setting with linear Markov decision processes. The bound in Kakade et al. (2020) depends on the dimension of a feature representation and is vacuous if process noise is absent, Croissant et al. (2024) relies on a combination of eluder dimension and covering numbers, Cohen et al. (2019); Abeille & Lazaric (2020) focus on the linear-quadratic regulator. More generally, the challenges of the optimism in the face of uncertainty paradigm lie in i) computing confidence sets and ii) optimizing over optimistic policies on the basis of these confidence sets. The suggested complexity measures, such as the eluder dimension (Foster & Rakhlin, 2023) and variants thereof, tend to be difficult to evaluate and to assert boundedness.

Our algorithms do not introduce optimism in the face of uncertainty, thereby avoiding the computation of confidence intervals and optimisitic policies (in Alg. 1 policies can even be computed offline). However, exploration must be incorporated and analyzed explicitly, and we rely on the notion of persistence of excitation to do so. Persistence of excitation has a rich history in system identification, statistics, and adaptive control (Ljung, 1999; Slotine & Li, 1991), and is tailored to continuous state-action systems, unlike eluder dimension, Bellman-rank, etc., which are rooted in (linear) Markov decision processes and dynamic programming. We further use packing numbers, which is standard and well-established, as a complexity measure for our function class. The overall policy-regret guarantees share a similar square root dependence on $N$ with state-of-the-art algorithms based on the principle of optimism, but feature a complexity measure suitable for continuous state-action systems.

**The separation principle** describes the algorithmic idea of separating identification and optimal control. This approach has turned out to be very effective for linear quadratic regulators (see, e.g., Dean et al., 2018; Mania et al., 2019; Simchowitz & Foster, 2020). The earlier work by Dean et al. (2018) incorporated model-uncertainty via a robust control synthesis; however it was later found (Mania et al., 2019; Simchowitz & Foster, 2020) that the robust control synthesis can be replaced by solving the standard linear quadratic regulator problem, thereby simplifying control synthesis. The works by Mania et al. (2019); Simchowitz & Foster (2020) offer improved policy-regret guarantees and lower bounds (Simchowitz & Foster, 2020).

This work shares the idea of certainty-equivalence with Mania et al. (2019); Simchowitz & Foster (2020), but with the key difference that model candidates are sampled from a posterior over models rather than the maximum a-posteriori. The posterior sampling enables a Hedge-type analysis that facilitates generalization beyond linear systems, whereas the works mentioned rely on perturbation bounds to discrete Riccati equations, which are tailored to linear time-invariant systems. We further recover a $\mathcal{O}(\sqrt{d_{\mathrm{u}}Np})$ policy-regret guarantee for linear systems matching Simchowitz & Foster (2020).

**Multi-model adaptive control** emphasizes the versatility of a system to handle diverse operating conditions by switching among multiple candidate models and associated controllers (Narendra & Balakrishnan, 1997; Anderson et al., 2000; Hespanha et al., 2001; Muehlebach, 2023; Chatzikiriakos & Iannelli, 2024). There is a supervisory policy that tracks the performance of the running controller and, if necessary, applies another more appropriate one based on a switching logic. Oftentimes the switching criterion follows the model (and the corresponding controller) with the smallest estimation error integral (Liberzon, 2003; Anderson & Dehghani, 2008) or implements performance-based falsification (Safonov & Tsao, 1997). These works mainly focus on asymptotic stabilization, whereas this article explores online reinforcement learning characterized by nonasymptotic policy-regret.

**Online control with switching policies** is closely related to adaptive control with multiple models. Nonetheless, instead of tackling asymptotic stabilization, online control addresses optimal control from a modern finite-sample perspective. Specifically, Li et al. (2023); Kim & Lavaei (2024) consider regulating a nonlinear dynamical system by iteratively selecting a control input from a finite set of candidate control policies. The key principles are to use the system trajectory driven by the chosen controller as a performance criterion to remove non-stabilizing controllers and identify the best stabilizing controller in hindsight via Exp3 (Auer et al., 2002), a classical multi-armed bandit algorithm. The regret bounds therein scale sublinearly with the time horizon, but grow exponentially with the number of non-stabilizing controllers. In contrast, in the setting with finite candidate models, our algorithm attains a favorable logarithmic regret in terms of both the time horizon and the number of models. Furthermore, we extend the design and analysis to handle a continuum of nonlinear candidate models contained in a bounded subset of a normed space. Our multi-model perspective is also connected to the line of works by Doya et al. (2002); Rajeswaran et al. (2017); Modi et al. (2020) that use dynamic convex combinations of an ensemble of models to synthesize policies. In contrast, we tackle a challenging non-episodic scenario without state reset and handle more general model classes including parametric families and families with infinite cardinality.

**Online optimization for continuous control** is based on online performance optimization over an appropriate policy class. A frequent policy class is given by disturbance-action policies (Agarwal et al., 2019; Hazan et al., 2020; Simchowitz et al., 2020; Li et al., 2021; Chen & Hazan, 2021; Zhao et al., 2023), which are motivated by the design of robust controllers for linear systems (Gartska & Wets, 1974; Löfberg, 2003; Goulart et al., 2006). These works are restricted to linear systems, although growing attention is currently paid to online nonlinear control, where additional structure,

e.g., matched uncertainty (Boffi et al., 2021)), contractive perturbation (Lin et al., 2024) or incremental input-to-state stability (Karapetyan et al., 2023) is introduced. Another approach is to parametrize policies, e.g., via linear functions and to apply zeroth-order or first-order optimization techniques (see, e.g., Fazel et al., 2018; Ma et al., 2024; Hu et al., 2023). We refer the readers to Hazan & Singh (2022); Tsiamis et al. (2023) for comprehensive reviews. Compared to the algorithms and analysis presented herein, the benchmarks in these works are typically different, and focus, for example, on the set of linear disturbance feedback policies or various policy parametrizations that are optimized subject to a fixed system model. The system dynamics are a-priori known to the decision-maker. Hence, no tradeoff between exploration and exploitation is needed and the methods are more closely related to online or stochastic optimization, rather than reinforcement learning.

In summary, all the aforementioned works provide a comprehensive ground for online decision-making. Nonetheless, achieving sublinear policy regret in an online non-episodic regime encompassing a broad class of nonlinear dynamics remains a critical challenge. In this article, we adopt a multi-model perspective and provide a suite of algorithms that identify the best candidate model and apply a certainty-equivalent policy, all equipped with nonasymptotic frequentist policy-regret guarantees. More precisely, for linear quadratic regulator problems we recover $\mathcal{O}(\sqrt{d_\mathrm{u} N p})$ regret bounds derived in earlier works (e.g., (Simchowitz & Foster, 2020)), but our analysis generalizes to nonlinear dynamics and smooth non-quadratic stage costs. In the nonlinear setting, prior work has established similar $\mathcal{O}(\sqrt{N})$ results, however, under different assumptions, such as parametrization via kernels (Kakade et al., 2020), dynamics linear in the parameters (Boffi et al., 2021), or contraction (Lin et al., 2024). Other works, such as Li et al. (2023) achieve policy regret of $\mathcal{O}(m^{1/3} N^{2/3}) + \exp(\mathcal{O}(|\mathcal{M}|))$ in a finite model setting ($|\mathcal{M}|$ is the number of potentially destabilizing candidate controllers), which is much worse than our result $\mathcal{O}(\ln(N) + \ln(m))$. While aligned with posterior sampling reinforcement learning (Osband et al., 2013), we tackle an online non-episodic setting with dependent continuous states, actions, and losses. We further establish frequentist policy-regret guarantees, closed-loop stability, and bounded losses. Compared to optimistic methods (Abeille & Lazaric, 2020; Croissant et al., 2024) that achieve $\mathcal{O}(\sqrt{N})$ regret and a square-root dependence on the eluder dimension, our assumptions are explicit and encompass a large set of nonlinear systems. Further, we characterize the scaling of policy regret with respect to complexity measures of the model class in the finite, nonparametric, and parametric regimes. Our approach avoids the computation of optimistic policies or confidence regions and can therefore be directly integrated in nonlinear model predictive control techniques (Rawlings et al., 2017; Borrelli et al., 2017). We envision that fruitful advances in these directions will further consolidate our multi-model perspective on online decision-making.

## B NUMERICAL EXAMPLE

We present results of two numerical simulations to illustrate our algorithms. The first simulation consists of linear time-invariant dynamics, and implements Alg. 1 and Alg. 2, whereas the second simulation is based on the swing-up of a nonlinear pendulum-on-a-cart system and implements Alg. 1. The first simulation focuses on the realizable setting, that is, the true dynamics are within the set of models, whereas in the second simulation the true dynamics are not contained in the set of candidate models. In all experiments the algorithms show rapid convergence to near-optimal steady state, while having a small computational footprint.

### B.1 LINEAR TIME-INVARIANT DYNAMICS

We consider first a linear time-invariant dynamical system of dimension $d_\mathrm{x} = 20$ and $d_\mathrm{u} = 5$ and apply the two algorithms Alg. 1 and Alg. 3. The stage cost is $l(x, u) = |x|^2 + |u|^2$. The dynamics $f$ (unknown to the decision-maker) consist of five four-dimensional leaky integrators of the type $x_{k+1}^i = 0.8 x_k^i + x_k^{i+1}$, $i = 1, \dots, 3$. The dynamics are relatively challenging for control, as there is a lag of five steps until a change in the input affects $x_k^1$. The above dynamics are compactly written as $f(x, u) = Ax + Bu$, where $x \in \mathbb{R}^{d_\mathrm{x}}$ is the state, $u \in \mathbb{R}^{d_\mathrm{u}}$ is the input, $A = I_5 \otimes A_0, B = I_5 \otimes B_0$ are system matrices, $I_5$ is an identity matrix of size 5, $\otimes$ denotes the Kronecker product, and

$$
A_0 = \begin{bmatrix} 0.8 & 1 & 0 & 0 \\ 0 & 0.8 & 1 & 0 \\ 0 & 0 & 0.8 & 1 \\ 0 & 0 & 0 & 0.8 \end{bmatrix}, \quad B_0 = \begin{bmatrix} 0 \\ 0 \\ 0 \\ 1 \end{bmatrix}.
$$

It is assumed that the elements of the matrices $A$ and $B$ that define the dynamics are unknown with respect to an absolute error of 0.1 and relative error of 20%, which gives rise to a large set of possible models including some open-loop unstable ones. For instance, the $jk$-th element $a_{jk}$ of $A$, is known to be in the range $[0.8a_{jk} - 0.1, 1.2a_{jk} + 0.1]$.

### B.1.1  SETTING S1

**Set-up:**  We generate $m$ candidate models $f^i(x, u) = A^i x + B^i u$ at random, whereby each element of $A^i$, $B^i$ is randomly drawn from the known parameter range, e.g., the $jk$-th element $a^i_{jk}$ of $A^i$ is sampled from the uniform distribution over $[0.8a_{jk} - 0.1, 1.2a_{jk} + 0.1]$. The feedback policy $\mu^i$ related to candidate model $f^i$ is

$$\mu^i(x) = -K^i x, \quad \text{where} \quad K^i = (I + B^{i\top} P B^i)^{-1} B^{i\top} P^i A^i,$$

and $P^i \in \mathbb{R}^{d_x \times d_x}$ is a positive definite matrix satisfying the discrete-time algebraic Riccati equation involving $A^i$ and $B^i$ Bertsekas (2017). The policies $\mu^i$ are computed through the built-in `dlqr` command in MATLAB and the settings of Alg. 1 were chosen as specified in Thm. 3.2, that is

$$\eta = 10, \quad \sigma^2_{uk} = \frac{2}{\eta M} \left( \frac{2}{\lceil k/M \rceil} + \frac{\ln(2m)}{\lceil k/M \rceil^2} \right), \quad M = 2.$$

**Assumptions of Thm. 3.2:**  Ass. 1-3 are clearly satisfied:

- The cost-to-go function $V$ is given by $V(x) = x^\top P x$, where $P$ satisfies the discrete-time algebraic Riccati equation involving $A$ and $B$. Hence, Ass. 1 is satisfied with $\gamma = \text{tr}(P)\sigma^2$, $L_u = \bar{\sigma}(P)$, where $\bar{\sigma}$ denotes the maximum singular value of a matrix.
- Ass. 2 is satisfied with $\underline{c}_V = 0$, $\underline{L}_V = \underline{\sigma}(P)$, where $\underline{\sigma}$ is the minimum singular value of a matrix.
- Ass. 3 is satisfied (for any $M > 0$) with

$$c_e = \min_{i \in \{1,\dots,m\}} |B^i - B|^2_F,$$

  for example, as can be seen from (14). Larger values of $c_e$ can be achieved when choosing $M$ larger and factoring in the controllability Gramian $W^c_k$.

Please note that the constants $c_e, \underline{c}_V, \underline{L}_V, \gamma$ only appear in the resulting policy-regret bounds and are not needed for running Alg. 1.

**Computational complexity:**  All experiments run on a Laptop (Intel Core i7 processor with 2.30GHz; 32 GB of random access memory) and are executed in a few minutes even when increasing the number of candidate model up to 10,000. The offline computation of the policies $\mu^i$ has cost $\mathcal{O}(d_x^3 + d_u^2 d_x)$, the online computation in Alg. 1 is $\mathcal{O}(d_x m + d_u d_x)$.

### B.1.2  SETTING S2

**Set-up:**  The parameter space $\Omega \subset \mathbb{R}^p$ with $p = d_x^2 + d_x d_u = 500$ covers the entire parameter range

$$\Omega = \{(\bar{A}, \bar{B}) \in \mathbb{R}^p \mid \bar{a}_{jk} \in [0.8a_{jk} - 0.1, 1.2a_{jk} + 0.1], \quad \bar{b}_{jk} \in [0.8b_{jk} - 0.1, 1.2b_{jk} + 0.1]\},$$

where we slightly abuse notation to avoid distinguishing between different ways of stacking vectors and matrices (we will frequently do so in the following as the stacking is clear from context). For a given set of matrices $(\bar{A}, \bar{B}) \in \Omega$ the corresponding feedback controller $\bar{\mu}(x) = -\bar{K}x$ is given as in setting S1 and requires solving the discrete-time algebraic Riccati equation involving $\bar{A}, \bar{B}$. As before, the feedback policies are computed through the built-in `dlqr` command in MATLAB and the settings of Alg. 3 are chosen as specified in Thm. E.1, that is,

$$\eta = 10, \quad \epsilon = \sqrt{p/N}, \quad \sigma^2_{uk} = \frac{2}{\eta M \epsilon} \left( \frac{2}{\lceil k/M \rceil} + \frac{p}{\lceil k/M \rceil^2} \right), \quad M = 5.$$

The posterior distribution over models in Alg. 3 is updated by a recursive least squares algorithm and we set $b \to \infty$. The recursive implementation has the advantage that reasonable estimates of $A$ and $B$ are already provided in the first $p$ steps, which is important for the initial transient behavior.

**Assumptions of Thm. E.1:** Ass. 2, Ass. 6, and Ass. 7 are satisfied:

- The cost-to-go function is given by $V(x) = x^\top P x$, where $P$ satisfies the discrete-time algebraic Riccati equation involving $A$ and $B$. One can easily show that Ass. 7 is satisfied by applying Prop. D.1. (As pointed out in (Simchowitz & Foster, 2020, Prop. 6), for example, the policies $\mu_\theta$ are continuously dependent on the system parameters $\theta = (A, B)$.)

- Ass. 7 is satisfied with $\underline{c}_{\mathrm{V}} = 0$, $\underline{L}_{\mathrm{V}} = \underline{\sigma}(P)$.

- Ass. 6 is satisfied in view of (14) in Sec. 3, provided that $M = 5$, which ensures that the controllability Gramian $W_k^{\mathrm{c}}$ is full rank for any feedback gain $\bar{K}$. (The dynamics $A, B$ give rise to decoupled four-dimensional leaky integrators, hence the Gramian $W_4^{\mathrm{c}}$ defined in Sec. 3 is guaranteed to be full rank.)

**Computational complexity:** Sampling the parameter $\theta_k$ in Alg. 3 amounts to sampling from a truncated Gaussian, where the mean and covariance of the Gaussian are computed via the recursive least squares algorithm. The computation of mean and covariance can be done in at most $\mathcal{O}(d_{\mathrm{x}}^2 + d_{\mathrm{x}} d_{\mathrm{u}})$ elementary operations at each iteration $k$. We then sample $\theta_k$ by applying rejection sampling (although much more efficient approaches could be applied). The policy $\mu_{\theta_k}$ is then computed by solving the corresponding discrete-time algebraic Riccati equation, which requires at most $\mathcal{O}(d_{\mathrm{x}}^3 + d_{\mathrm{u}}^2 d_{\mathrm{x}})$ elementary operations.

### B.1.3 RESULTS

Simulation results for the setting `S1` are shown in Fig. 1, whereas the results for setting `S2` are shown in Fig. 2. A rapid convergence to near-optimal steady-state behavior can be observed in both cases. We note that the space of parameters in Alg. 3 is uncountable compared to Alg. 1 and therefore Alg. 3 takes about twice as long to converge. Alg. 1 achieves optimal steady-state performance very quickly (in about twenty steps). We therefore believe that Alg. 1 provides an algorithmic paradigm that is applicable to many emerging real-world machine learning and engineering challenges, including the control of intelligent transportation systems or automated supply chains.

To showcase the scalability of our algorithms, we provide comparison results when the number of models $m$ in Alg. 1 is increased from 10 to $10,000$. We perform 40 independent realizations of Alg. 1 for each value of $m$ and show the corresponding policy regret in Fig. 3a (averaged over the 40 realizations). Once again we observe a fast initial transient phase after which the policy regret stabilizes and near-optimal steady-state performance is achieved. Fig. 3b shows the corresponding evolution of the two-norm of the state trajectory on a single realization. The plots highlight that Alg. 1 scales favorably in the number of models.

### B.2 SWING-UP OF A NONLINEAR PENDULUM-ON-A-CART

We showcase that our algorithms work seamlessly with nonlinear systems.

**Set-up:** We consider the following nonlinear continuous-time pendulum-on-a-cart system dynamics:

$$\ddot{r}(t) = \frac{\frac{\bar{u}(t)}{m_{\mathrm{p}}} - (g\sin(\varphi(t)) - d_2\dot{\varphi}(t))\cos(\varphi(t)) + l_{\mathrm{p}}\dot{\varphi}(t)^2\sin(\varphi(t))}{\frac{m_{\mathrm{c}}}{m_{\mathrm{p}}} + \sin(\varphi(t))^2},$$

$$\ddot{\varphi}(t) = \frac{-\cos(\varphi(t))\frac{\bar{u}(t)}{m_{\mathrm{p}}} + \frac{m_{\mathrm{c}}+m_{\mathrm{p}}}{m_{\mathrm{p}}}(g\sin(\varphi(t)) - d_2\dot{\varphi}(t)) - l_{\mathrm{p}}\dot{\varphi}(t)^2\sin\varphi(t)\cos\varphi(t)}{l_{\mathrm{p}}(\frac{m_{\mathrm{c}}}{m_{\mathrm{p}}} + \sin(\varphi(t))^2)},$$

$$\bar{u}(t) = \max(\min(u(t), F_{\max}), -F_{\max}) - d_1\dot{r}(t),$$

where $r(t)$ denotes the position of the cart on the rail and $\varphi(t)$ the pendulum angle ($\varphi = 0$ corresponds to the upright equilibrium), and parameters $m_{\mathrm{c}}$, $m_{\mathrm{p}}$, $l_{\mathrm{p}}$, $g$, $d_1$, $d_2$, and $F_{\max}$, corresponding to cart mass, pendulum mass, pendulum length, gravitational constant, cart damping, pendulum damping, and force limit, respectively. The rail has a finite length of 1m, that is $r(t) \in [-0.5, 0.5]$, which is incorporated by modifying the above dynamics accordingly for $r(t) \in \{-0.5, 0.5\}$. The continuous-time dynamics are discretized with the classical Runge-Kutta method with step size 0.02. The nominal parameter values, that correspond to an actual pendulum-on-a-cart system are given as $m_{\mathrm{c}} = 1.73\mathrm{kg}$,

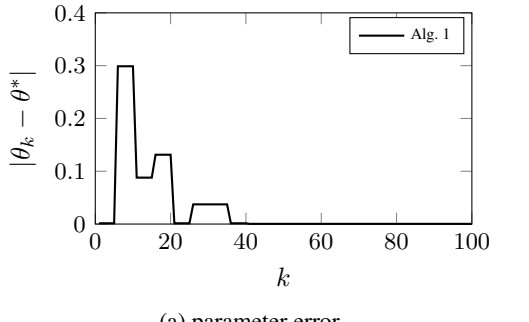
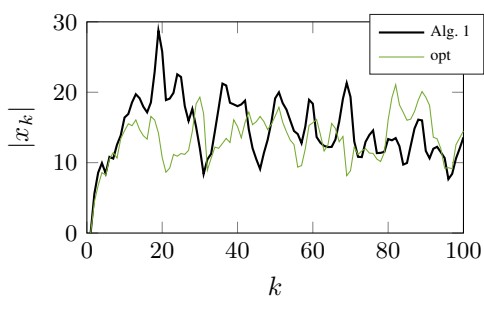

(a) parameter error

(b) two-norm of state trajectory

Figure 1: The first panel shows the evolution of the parameter error of Alg. 1, while the second panel shows the evolution of the two-norm of the states. The green line indicates the performance of the optimal (steady-state) policy on a different realization of $n_k$. We note that near-optimal steady state performance is reached in about 25 steps.

$m_\mathrm{p} = 0.175$kg, $l_\mathrm{p} = 0.28$m, $F_\mathrm{max} = 15$N, $d_1 = d_2 = 0$, and $g = 9.81$m/s$^2$. We generate $m = 500$ candidate models by randomly sampling the parameters $m_\mathrm{c}, m_\mathrm{p}, l_\mathrm{p}$ within $\pm 20\%$ of their nominal values and $d_1, d_2$ uniformly in the range $[0, 1]$ and $[0, 0.1]$ respectively. We further emphasize that the true dynamics with nominal parameter values are not contained in the set of candidate models. The stage cost is $1 - \cos(\varphi(t)) + 0.01u(t)^2 + 0.1r(t)^2$, and process noise is incorporated by perturbing the input $u$ with a sequence of independent Gaussian random variables sampled at $0.02s$.

**Control design:** The feedback policies arise from a model-predictive controller. More precisely, the policy $\mu^i$ is computed by solving a numerical optimal control problem that minimizes the stage cost over a horizon of 50 steps with a Runge-Kutta discretization of the dynamics corresponding to model $i$ with step size 0.04. The numerical optimal control problem includes state constraints due to the finite rail length and input constraints, and is solved with IPOPT (Wächter & Biegler, 2006) interfaced through CasADi (Andersson et al., 2018).

**Learning parameters:** The settings of Alg. 1 are chosen as specified in Thm. 3.2, that is,

$$\eta = 10, \quad \sigma_{\mathrm{u}k}^2 = \frac{10}{\eta M} \left( \frac{2}{\lceil k/M \rceil} + \frac{\ln(2m)}{\lceil k/M \rceil^2} \right), \quad M = 5.$$

**Computational complexity:** All experiments run on a Laptop (Intel Core i7 processor with 2.30GHz; 32GB of random access memory) and are executed in less than two minutes. The algorithms are implemented in MATLAB and have not been optimized. The bottleneck in the execution is the numerical solution of the optimal control problem at every iteration, which is independent of the number of candidate models.

### B.2.1 RESULTS

Simulation results for the setting `S1` are shown in Fig. 4-Fig. 5 and highlight that the pendulum is seamlessly brought to upright position while taking input and state constraints into account. Compared to the ground-truth swing-up, based on the true dynamics, the swing-up takes about 40 steps more ($\approx$ 0.8s), due to the model identification. Even though the true dynamics are not included in the set of candidate models, the model-estimation converges in about 100 steps. The fact that there are multiple models that approximate the underlying dynamics well (see Fig. 5b) does not pose any issues, and the pendulum can be easily balanced in upright position. Moreover, the online learning converges quickly (100 steps), despite the fact that there is a large variability in the set of candidate models ($m = 500$). At every iteration only a single feedback policy needs to be evaluated, which amounts to solving a numerical optimal control problem with input and state constraints. No difficulties were observed in the optimization and a single iteration of Alg. 1 (including policy evaluation) is carried out in about 60ms, even though the implementation is not optimized for speed.

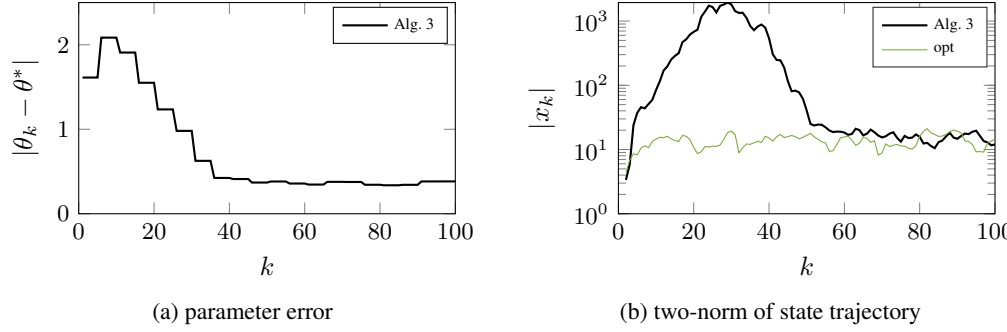

(a) parameter error

(b) two-norm of state trajectory

Figure 2: The first panel shows the evolution of the parameter error of Alg. 3, while the second panel shows the evolution of the two-norm of the states. The green line indicates the performance of the optimal (steady-state) policy on a different realization of $n_k$. Compared to Fig. 1b the overshoot is larger and the convergence to near-optimal performance requires about 60 iterations.

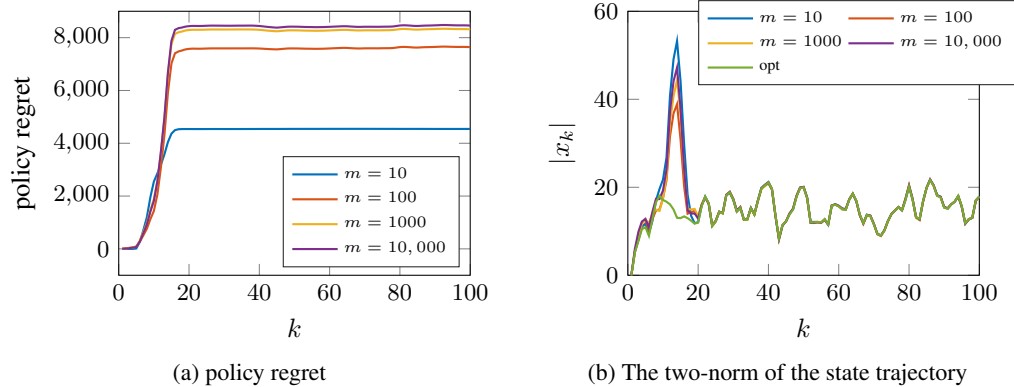

(a) policy regret

(b) The two-norm of the state trajectory

Figure 3: The first panel shows the change in policy regret of Alg. 1 when varying $m$. The second panel shows the evolution of the two-norm of the state trajectory. We note that the behavior is consistent over the different values of $m$ (from 10 to 10,000, which amounts to three orders of magnitude). The green line indicates the performance of the optimal (steady-state) policy on the *same* realization of $n_k$.

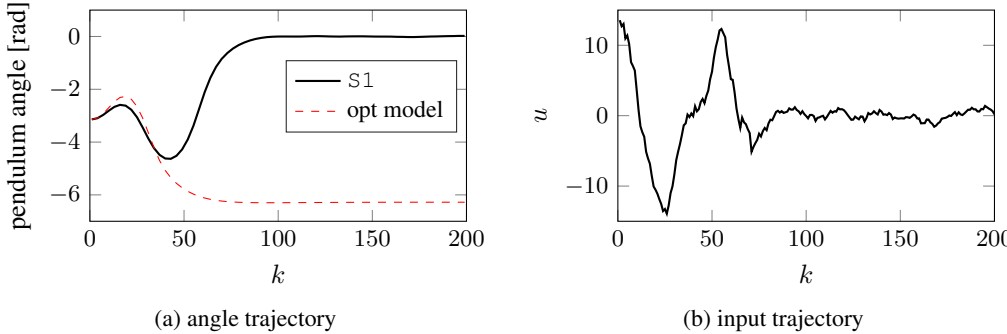

(a) angle trajectory

(b) input trajectory

Figure 4: The first panel shows the evolution of pendulum angle when running Alg. 1 (solid black) and the pendulum angle arising from a swing-up based on the true dynamics (red dashed). The second panel shows the evolution of input when swinging up the pendulum. Alg. 1 performs the swing-up after around 80 iterations, whereas with the true dynamics the swing-up requires about 60 iterations.

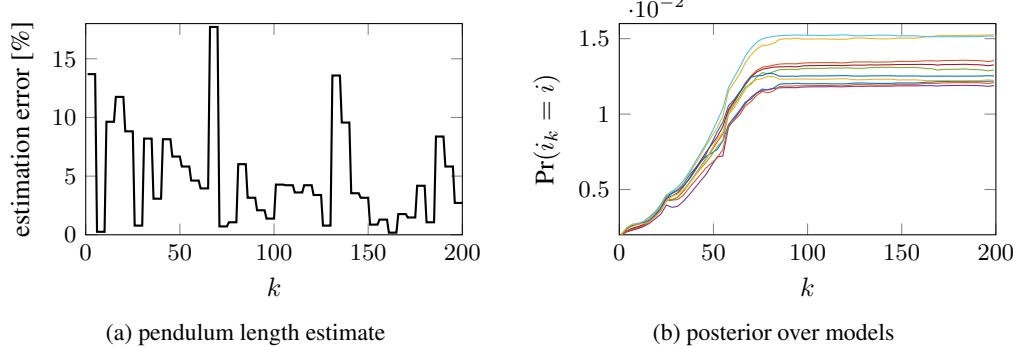

(a) pendulum length estimate                (b) posterior over models

Figure 5: The first panel shows the evolution of the pendulum length estimate when running Alg. 1, whereas the second panel shows the evolution of posterior over models. *For the sake of visualization only the ten most likely models are shown, even though $m = 500$.* The true dynamics are not included in the set of candidate models. Nonetheless the pendulum length is rapidly estimated to an accuracy below 5% and the posterior over models converges after about 80 steps.

## C  DETAILS OF SEC. 3.1

We first state and prove two intermediate lemmas that are used in the proof of Prop. 3.1. The two lemmas express the fact that the larger the expected model deviation $f^i - f$ (accumulated over the past steps), the smaller the corresponding probability of selecting model $i$.

**Lemma C.1** *For any step size $\eta > 0$ it holds that*

$$\Pr(i_k = i) \le \mathrm{E}[e^{-\eta(s_k^i - s_k^j)}],$$

*for all $s_k^j$ (and in particular for $s_k^j = s_k^*$ corresponding to $f$).*

**Proof** We note that $p_k^i$ is given by

$$p_k^i = \frac{e^{-\eta s_k^i}}{\sum_{j=1}^m e^{-\eta s_k^j}} \le e^{-\eta(s_k^i - \bar{s}_k)} \le e^{-\eta(s_k^i - s_k^j)},$$

for all $j \in \{1, \dots, m\}$, where $\bar{s}_k = \min_{i \in \{1,\dots,m\}} s_k^i$. In addition, it holds that

$$\Pr(i_k = i) = \mathrm{E}[\mathbb{1}_{i_k = i}] = \mathrm{E}[\mathrm{E}[\mathbb{1}_{i_k = i} | x_1, \dots, x_k, u_k, n_k]] = \mathrm{E}[p_k^i],$$

where $\mathbb{1}$ denotes the indicator function, which yields the desired result. $\qquad\square$

**Lemma C.2** *Let*

$$l_k^i := \frac{|x_{k+1} - f^i(x_k, u_k)|^2}{1 + |(x_k, u_k)|^2 / b^2},$$

*where $b > 0$ is constant. Let $\mathcal{F}_k$ denote the collection of random variables $x_j, u_j, i_j, n_{j-1}, n_{\mathrm{u}j}$ up to time $k$. Then, the following bound holds for all $0 < \eta \le \min\{1/(4\sigma^2), 1/(2L^2 b^2)\}$ and for all $1 \le q \le k$:*

$$\mathrm{E}[e^{-\eta(l_k^i - l_k^*)} | \mathcal{F}_q] \le \exp\left(-\frac{\eta}{4} \mathrm{E}\left[\frac{|f - f^i|^2}{1 + |(x_k, u_k)|^2 / b^2} \Big| \mathcal{F}_q\right]\right),$$

*where $f$ stands for $f(x_k, u_k)$ and $f^i$ for $f^i(x_k, u_k)$, and where $l_k^*$ corresponds to the loss of the candidate $f$.*

**Proof** We note that $|x_{k+1} - f^i(x_k, u_k)|^2$ can be expressed as

$$|f - f^i + n_k|^2 = |f - f^i|^2 + 2n_k^\top(f - f^i) + n_k^\top n_k,$$

and, as a result, $l_k^i - l_k^*$ is given by

$$\frac{|f - f^i|^2 + 2n_k^\top(f - f^i)}{1 + |(x_k, u_k)|^2 / b^2}.$$

Hence, conditioned on $x_k, u_k$, the randomness in $l_k^i - l_k^*$ is solely due to $n_k^\top (f - f^i)$, which describes a sum of $d_\mathrm{x}$ independent Gaussian random variables, weighted by the components of $f - f^i$. As a result, we exploit the closed-form expression for the moment generating function of a Gaussian, which yields

$$\mathrm{E}[e^{-\eta(l_k^i - l_k^*)}|x_k, u_k] \leq \exp\left( \frac{2\eta^2\sigma^2|f - f^i|^2}{1 + |(x_k, u_k)|^2/b^2} - \eta\frac{|f - f^i|^2}{1 + |(x_k, u_k)|^2/b^2} \right).$$

Thus, for $\eta \leq 1/(4\sigma^2)$, the following bound holds

$$\mathrm{E}[e^{-\eta(l_k^i - l_k^*)}|\mathcal{F}_q] \leq \mathrm{E}\left[ \exp\left( -\frac{\eta|f - f^i|^2/2}{1 + |(x_k, u_k)|^2/b^2} \right)\Big|\mathcal{F}_q \right].$$

As a result of the Lipschitz continuity of $f$ and $f^i$, the term

$$0 \leq \frac{|f - f^i|^2}{1 + |(x_k, u_k)|^2/b^2} \leq 4L^2b^2 \tag{7}$$

is bounded. When deriving the previous inequality we used the fact that $|f^i(x_k, u_k) - f(x_k, u_k)| \leq |f^i(x_k, u_k) - f^i(0,0)| + |f(x_k, u_k) - f(0,0)| \leq 2L|(x_k, u_k)|$ by Lipschitz continuity of $f$ and $f^i$.[3] We can therefore apply a "Poissonian" inequality (see, e.g., Cesa-Bianchi & Lugosi, 2006, App. A), which yields

$$\mathrm{E}[e^{-\eta(l_k^i - l_k^*)}|\mathcal{F}_q] \leq \exp\left( \frac{(e^{-2\eta L^2b^2} - 1)}{4L^2b^2}\mathrm{E}\left[ \frac{|f - f^i|^2}{1 + |(x_k, u_k)|^2/b^2}\Big|\mathcal{F}_q \right] \right),$$

for all $\eta \leq 1/(4\sigma^2)$. The desired result follows from the fact that $(e^{-2\eta L^2b^2} - 1)/(4L^2b^2) \leq -\eta/4$ for all $\eta \leq \min\{1/(4\sigma^2), 1/(2L^2b^2)\}$. □

### C.1 Proof of Prop. 3.1

We first consider the iterations $k = k'M + 1$, for $k' = 0, 1, \ldots$. These are the iterations $k$ where the random variable $i_k$ is updated according to the distribution $p_k^i$ (conditional on $x_k, u_k$). It will be useful to introduce the variables $\bar{l}_{k'}^i$ as follows:

$$\bar{l}_{k'}^i = \sum_{j=1}^M l_{k'M+j}^i,$$

which corresponds to a sum of the variables $l_k^i$ over $M$ steps. Let $\mathcal{F}_{k'}$ denote the collection of all random variables $(x_k, u_k, i_k, n_{k-1}, n_{\mathrm{u}k})$ up to time $k = k'M + 1$. We condition on $\mathcal{F}_{k'-1}$ and conclude from Lemma C.2

$$\mathrm{E}[e^{-\eta(\bar{l}_{k'}^i - \bar{l}_{k'}^*)}|\mathcal{F}_{k'-1}] = \mathrm{E}[e^{-\eta\sum_{j=1}^M (l_{k'M+j}^i - l_{k'M+j}^*)}|\mathcal{F}_{k'-1}]$$

$$\leq \prod_{j=1}^M (\mathrm{E}[e^{-\eta M(l_{k'M+j}^i - l_{k'M+j}^*)}|\mathcal{F}_{k'-1}])^{1/M}$$

$$\leq \prod_{j=1}^M (e^{-\frac{\eta M}{4}\mathrm{E}[\frac{|f - f^i|^2}{1 + |(x_k, u_k)|^2/b^2}|\mathcal{F}_{k'-1}]})^{1/M}$$

$$\leq \exp\left( -\frac{\eta M\Delta}{4}\sigma_{\mathrm{u}(k-1)}^2 \right), \tag{8}$$

where we have used Hölder's inequality for the first inequality, Lemma C.2 for the second inequality, and Ass. 3 for the third inequality. As a result, by unrolling the recursion for $k' - 1, k' - 2, \ldots$, we conclude that

$$\mathrm{E}[e^{-\eta(s_k^i - s_k^*)}] \leq \exp\left( -\frac{\eta M\Delta}{4}\sum_{j=1}^{k'} \sigma_{\mathrm{u}(Mj)}^2 \right).$$

---

[3]We stated the inequality assuming $f(0,0) = f^i(0,0) = 0$. In the more general situation the upper bound $2|f^i(0,0) - f(0,0)|^2 + 8L^2b^2$ applies in (7).

By virtue of Lemma C.1, this implies

$$\Pr(i_k = i) \leq \exp\left(-\frac{\eta M \Delta}{4} \sum_{j=1}^{k'} \sigma^2_{\mathrm{u}(Mj)}\right).$$

The bound holds in fact also for $k + 1, k + 2$, until $k + (M - 1)$, since, by definition, $i_k = i_{k+1} = \cdots = i_{k+(M-1)}$. This proves the first bound of Prop. 3.1.

It remains to derive the second bound, which is done by approximating the sum over $\sigma^2_{\mathrm{u}k}$ from below. We find

$$\frac{\eta M \Delta}{4} \sum_{j=1}^{k'} \sigma^2_{\mathrm{u}(Mj)} = \sum_{j=1}^{k'}\left(\frac{2}{j} + \frac{\ln(m)}{j^2}\right) \geq \int_1^{k'} \frac{2}{j}\,\mathrm{d}j + \ln(m) \geq 2\ln(k') + \ln(m),$$

for $k' \geq 1$. This concludes that $\Pr(i_k = i) \leq 1/(mk'^2)$, due to the fact that $m \geq 1$. We further note that $k' = (k - 1)/M$ by our choice of $k$. However, $i_k$ remains unchanged for the $M$ next iterations, and therefore

$$\Pr(i_{k+M-1} = i) \leq \frac{M^2}{m(k-1)^2},$$

which holds for all $k \geq 2$ and $i \neq i^*$. This implies $\Pr(i_k = i) \leq M^2/(m(k - M)^2)$ for all $k \geq M + 1$ by a change of variables. Applying a union bound yields the second inequality of Prop. 3.1, i.e.,

$$\Pr(i_k \neq i^*) \leq \sum_{i \neq i^*} \Pr(i_k = i) \leq \frac{M^2}{(k-M)^2}.$$

$\square$

## C.2 PROOF OF THM. 3.2

We will use $V$ as a Lyapunov function and have

$$\mathrm{E}[V(x_{k+1})] = \mathrm{E}[V(x_{k+1})|i_k \neq i^*]\Pr(i_k \neq i^*) + \mathrm{E}[V(x_{k+1})|i_k = i^*]\Pr(i_k = i^*). \quad (9)$$

The first term can be further simplified in view of Lemma C.3, which yields

$$\mathrm{E}[V(x_{k+1})|x_k, i_k \neq i^*] \leq c_2 V(x_k) + \bar{L}_{\mathrm{V}} d_{\mathrm{x}} \sigma^2/2 + c_{\mathrm{o}}$$
$$- \mathrm{E}[l(x, u_k)|x_k, i_k \neq i^*] + (\bar{L}_{\mathrm{V}} L^2 + \bar{L}_{\mathrm{l}})d_{\mathrm{u}}\sigma^2_{\mathrm{u}k}. \quad (10)$$

The second term in (9) is bounded as a result of the Bellman-type inequality (4) for the policy $\mu$ (the policy that corresponds to $V$). It will be convenient to rewrite the bound (4) in the following way:

$$\mathrm{E}[V(f(x, u) + n)] \leq V(x) - \mathrm{E}[l(x, u)] + q(x) + d_{\mathrm{u}} L_{\mathrm{u}} \sigma^2_{\mathrm{u}},$$

where $u = \mu(x) + n_{\mathrm{u}}$, $(n_{\mathrm{u}}, n)$ are independent with $n_{\mathrm{u}} \sim \mathcal{N}(0, \sigma^2_{\mathrm{u}} I)$, $n \sim \mathcal{N}(0, \sigma^2 I)$, and $q(x)$ is chosen such that $q(x) \leq \gamma$ and $-\mathrm{E}[l(x, u)] + q(x) \leq 0$.[4] The function $q(x)$ is introduced to account for the fact that the policy $\mu$ might in principle also achieve a running cost $\mathrm{E}[l(x, u)] \leq \gamma$ in the short term, since $\gamma$ captures only the steady-state performance. As a result, we obtain

$$\mathrm{E}[V(x_{k+1})|i_k = i^*] \leq -\mathrm{E}[l(x_k, u_k)|i_k = i^*] + \mathrm{E}[V(x_k)] + \gamma_k + d_{\mathrm{u}} L_{\mathrm{u}} \sigma^2_{\mathrm{u}k}, \quad (11)$$

where $\gamma_k := \mathrm{E}[q(x_k)]$. By combining (10) and (11) with (9) we arrive at

$$\mathrm{E}[V(x_{k+1})] \leq \mathrm{E}[V(x_k)](c_2 \Pr(i_k \neq i^*) + 1) + \gamma_k - \mathrm{E}[l(x_k, u_k)]$$
$$+ \bar{L}_{\mathrm{u}} d_{\mathrm{u}} \sigma^2_{\mathrm{u}k} + (\bar{L}_{\mathrm{V}} d_{\mathrm{x}} \sigma^2/2 + c_{\mathrm{o}})\Pr(i_k \neq i^*),$$

where $\bar{L}_{\mathrm{u}} := \bar{L}_{\mathrm{V}} L^2 + \bar{L}_{\mathrm{l}} + L_{\mathrm{u}}$. As a result of Prop. 3.1, we know that $\Pr(i_k \neq i^*) \leq M^2/(k - M)^2$ for $k \geq M + 1$. We further note that $\mathrm{E}[l(x_k, u_k)] \geq \gamma_k$ and $\gamma_k \leq \gamma$ (by our choice of $\gamma_k$). We now invoke Lemma C.4 and conclude

$$\sum_{k=1}^{N}(\mathrm{E}[l(x_k, u_k)] - \gamma_k) \leq c_\alpha \bar{L}_{\mathrm{u}} d_{\mathrm{u}} \sum_{k=1}^{N} \sigma^2_{\mathrm{u}k} + c_\alpha(\bar{L}_{\mathrm{V}} d_{\mathrm{x}} \sigma^2/2 + c_{\mathrm{o}}) \sum_{k=1}^{N} \Pr(i_k \neq i^*),$$

---

[4]This can by achieved by setting $q(x) = \mathrm{E}[V(f(x, \mu(x) + n_{\mathrm{u}}) + n)] - V(x) + \mathrm{E}[l(x, \mu(x) + n_{\mathrm{u}})] - d_{\mathrm{u}}\sigma^2_{\mathrm{u}} L_{\mathrm{u}}$ for $\gamma \geq \mathrm{E}[l(x, \mu(x) + n_{\mathrm{u}})]$ and $q(x) = \gamma$ otherwise.

where we have used the fact that $V(x_1) = 0$ and the following calculation

$$\prod_{k=1}^{\infty}(c_2\mathrm{Pr}(i_k \neq i^*) + 1) \leq e^{c_2 \sum_{k=1}^{\infty} \mathrm{Pr}(i_k \neq i^*)} \leq e^{3Mc_2} = c_\alpha,$$

due to the fact that

$$\sum_{k=1}^{\infty}\mathrm{Pr}(i_k \neq i^*) \leq 2M - 1 + \sum_{k=2M}^{\infty}\frac{M^2}{(k - M)^2}$$

$$\leq 2M + \int_{2M}^{\infty}\frac{M^2}{(k - M)^2}\mathrm{d}k \leq 3M.$$

Moreover, we bound the sum over $\sigma_{\mathrm{u}k}^2$ as follows

$$\sum_{k=1}^{N}\sigma_{\mathrm{u}k}^2 = \frac{4}{\eta\Delta}\sum_{k=1}^{N}\left(\frac{2}{M\lceil k/M\rceil} + \frac{M\ln(m)}{(M\lceil k/M\rceil)^2}\right)$$

$$\leq \frac{4}{\eta\Delta}\sum_{k=1}^{N}\left(\frac{2}{k} + \frac{M\ln(m)}{k^2}\right)$$

$$\leq \frac{8}{\eta\Delta}(1 + \ln(N - 1) + M\ln(m)).$$

Combining the previous inequalities and taking advantage of the fact that $\gamma_k \leq \gamma$ yields the desired result. $\qquad\square$

### C.3 SUPPORTING LEMMAS IN THE PROOF OF THM. 3.2

This section contains two lemmas that support the proof of Thm. 3.2.

**Lemma C.3** *Let Ass. 2 be satisfied. Then, there exist two constants $c_2, c_\mathrm{o} \geq 0$ such that*

$$\mathrm{E}[V(f(x,\mu^i(x)+n_\mathrm{u})+n)] \leq c_2 V(x) - \mathrm{E}[l(x, \mu^i(x)+n_\mathrm{u})] + c_\mathrm{o} + (\bar{L}_\mathrm{V}L^2 + \bar{L}_\mathrm{l})d_\mathrm{u}\sigma_\mathrm{u}^2 + \frac{\bar{L}_\mathrm{V}}{2}d_\mathrm{x}\sigma^2,$$

*for all $x \in \mathbb{R}^{d_\mathrm{x}}$, $\sigma_\mathrm{u} > 0$, and $i \in \{1, \ldots, m\}$, where the constant $c_2$ is given by*

$$c_2 = (8L^2\bar{L}_\mathrm{V}(1 + L_\mu)^2 + 2\bar{L}_\mathrm{l}(1 + 2L_\mu^2))/\underline{L}_\mathrm{V},$$

*and $c_\mathrm{o}$ can be expressed as an explicit function of $\max_{i\in[m]}|\mu^i(0)|$, $V(0)$, $|\nabla V(0)|$, $l(0,0)$, $|\nabla l(0,0)|$, $|f(0,\mu(0))|$, and $\underline{c}_\mathrm{V}$. The random variables $n_\mathrm{u}$ and $n$ are independent and satisfy $n \sim \mathcal{N}(0, \sigma^2 I)$, $n_\mathrm{u} \sim \mathcal{N}(0, \sigma_\mathrm{u}^2 I)$.*

**Proof** We exploit smoothness of $V$ to bound $\mathrm{E}[V(f(x, \mu^i(x) + n_\mathrm{u}) + n)]$ by

$$\mathrm{E}[V(f(x, \mu^i(x) + n_\mathrm{u}))] + \frac{\bar{L}_\mathrm{V}}{2}d_\mathrm{x}\sigma^2,$$

where we used the fact that the term linear in $n$ vanishes in expectation. We further note that the term $V(f(x, \mu^i(x) + n_\mathrm{u}))$ can be bounded in a similar way:

$$V(f(x, \mu^i(x) + n_\mathrm{u})) \leq V(f(x, \mu^i(x))) + \nabla V(f(x, \mu^i(x)))^\top \nabla_u f(\xi)n_\mathrm{u} + \frac{\bar{L}_\mathrm{V}}{2}|\nabla_u f(\xi)n_\mathrm{u}|^2,$$

where we applied the mean value theorem to rewrite $f(x, \mu^i(x) + n_\mathrm{u}) - f(x, \mu^i(x))$ as $\nabla_u f(\xi)n_\mathrm{u}$ for some $\xi$ (dependent on $n_\mathrm{u}$). By applying Young's inequality and taking advantage of the fact that $\nabla_u f$ is bounded above we arrive at

$$\mathrm{E}[V(f(x, \mu^i(x) + n_\mathrm{u}))] \leq V(f(x, \mu^i(x))) + \frac{1}{2\bar{L}_\mathrm{V}}|\nabla V(f(x, \mu^i(x)))|^2 + \bar{L}_\mathrm{V}L^2d_\mathrm{u}\sigma_\mathrm{u}^2.$$

Due to smoothness, $V$ is guaranteed to satisfy

$$|\nabla V(x)| \leq c_{\mathrm{o}1} + \bar{L}_\mathrm{V}|x|, \quad V(x) \leq c_{\mathrm{o}2} + \bar{L}_\mathrm{V}|x|^2,$$

where $c_{\mathrm{o1}} = |\nabla V(0)|$, and the constant $c_{\mathrm{o2}} \geq 0$ is similarly related to $|\nabla V(0)|$ and $V(0)$. As a result, we obtain the following upper bound on $\mathrm{E}[V(f(x, \mu^i(x) + n_{\mathrm{u}}))]$:

$$\mathrm{E}[V(f(x, \mu^i(x) + n_{\mathrm{u}}))] \leq c_{\mathrm{o2}} + \frac{c_{\mathrm{o1}}^2}{\bar{L}_{\mathrm{V}}} + 2\bar{L}_{\mathrm{V}}|f(x, \mu^i(x))|^2 + \bar{L}_{\mathrm{V}}L^2 d_{\mathrm{u}}\sigma_{\mathrm{u}}^2.$$

The fact that $f(x, \mu^i(x))$ is $L(1 + L_\mu)$ Lipschitz can be used to conclude that $|f(x, \mu^i(x))|^2 \leq c_{\mathrm{o3}} + 2L^2(1 + L_\mu)^2|x|^2$, where $c_{\mathrm{o3}} = 2|f(0, \mu(0))|^2$, which, in turn, yields the following upper bound

$$\mathrm{E}[V(f(x, \mu^i(x) + n_{\mathrm{u}}))] \leq c_{\mathrm{o2}} + \frac{c_{\mathrm{o1}}^2}{\bar{L}_{\mathrm{V}}} + 2\bar{L}_{\mathrm{V}}c_{\mathrm{o3}} + 4\bar{L}_{\mathrm{V}}L^2(1 + L_\mu)^2|x|^2 + \bar{L}_{\mathrm{V}}L^2 d_{\mathrm{u}}\sigma_{\mathrm{u}}^2.$$

We further note that the fact that $l$ is $\bar{L}_{\mathrm{l}}$ smooth and $l(x, u) \geq 0$ implies $l(x, u) \leq c_{\mathrm{o4}} + \bar{L}_{\mathrm{l}}(|x|^2 + |u|^2)$ and therefore

$$\mathrm{E}[l(x, \mu^i(x) + n_{\mathrm{u}})] \leq c_{\mathrm{o5}} + \bar{L}_{\mathrm{l}}d_{\mathrm{u}}\sigma_{\mathrm{u}}^2 + \bar{L}_{\mathrm{l}}(1 + 2L_\mu^2)|x|^2,$$

where $c_{\mathrm{o5}} \geq 0$ is related to $\max_{i \in [m]} |\mu^i(0)|$, $l(0, 0)$, and $|\nabla l(0, 0)|$. Combining the previous two inequalities results in

$$\mathrm{E}[V(f(x, \mu^i(x) + n_{\mathrm{u}}))] \leq c_{\mathrm{o6}} + \frac{c_2 L_{\mathrm{V}}}{2}|x|^2 - \mathrm{E}[l(x, \mu^i(x) + n_{\mathrm{u}})] + (\bar{L}_{\mathrm{V}}L^2 + \bar{L}_{\mathrm{l}})d_{\mathrm{u}}\sigma_{\mathrm{u}}^2,$$

where $c_{\mathrm{o6}} \geq 0$ is constant and can be expressed as a function of $\max_{i \in [m]} |\mu^i(0)|$, $V(0)$, $|\nabla V(0)|$, $l(0, 0)$, $|\nabla l(0, 0)|$, and $|f(0, \mu(0))|$. The result follows by inserting $\underline{L}_{\mathrm{V}}|x|^2/2 \leq V(x) + \underline{c}_V$ in the previous inequality. $\qquad \square$

**Lemma C.4** *Let the sequence*

$$V_{k+1} \leq (1 + \alpha_k)V_k + g_k^+ - g_k^-, \quad V_k \geq 0,$$

*be given, where $k = 1, 2, \ldots$, $\alpha_k \geq 0$ $g_k^+ \geq 0$, and $g_k^- \geq 0$ are arbitrary sequences such that $c_\alpha := \prod_{k=1}^\infty (1 + \alpha_k) < \infty$. Then, the following holds for all $N \geq 1$*

$$\sum_{j=1}^N g_j^- \leq c_\alpha \left( \sum_{j=1}^N g_j^+ + V_1 \right).$$

**Proof** By unrolling the linear difference equation we obtain

$$V_{N+1} \leq \prod_{k=1}^N (1 + \alpha_k)V_1 + \sum_{i=1}^N (g_i^+ - g_i^-) \prod_{j=i+1}^N (1 + \alpha_j)$$

$$\leq c_\alpha \left( V_1 + \sum_{i=1}^N g_i^+ \right) - \sum_{i=1}^N g_i^-,$$

where we exploited the fact that $\prod_{k=1}^N (1 + \alpha_k) < c_\alpha < \infty$. $\qquad \square$

## C.4 FINITE SECOND MOMENT

**Corollary C.5** *Let Ass. 2 be satisfied, let $\sigma_{\mathrm{uk}}^2$ be as in Prop. 3.1, let $l(x, u) \geq \underline{L}_{\mathrm{l}}|x|^2/2$ for some constant $\underline{L}_{\mathrm{l}} > 0$, and $\eta \leq \min\{1/(4M\sigma^2), 1/(2ML^2b^2)\}$. Let Ass. 3 be satisfied for at least the first*

$$k_0 := \left\lceil M \left( 1 + \sqrt{2\bar{L}_{\mathrm{V}}c_2/\underline{L}_{\mathrm{l}}} \right) \right\rceil$$

*steps. Then, it holds that*

$$\mathrm{E}[V(x_k)] \leq \max\{c_3, c_4\}, \quad \forall k \geq 1,$$

*with*

$$c_3 = c_2^{k_0} k_0 (\bar{L}_{\mathrm{V}} d_{\mathrm{x}} \sigma^2 + c_{\mathrm{o}} + (\bar{L}_{\mathrm{V}}L^2 + \bar{L}_{\mathrm{l}})d_{\mathrm{u}}\sigma_{\mathrm{u1}}^2),$$

$$c_4 = \frac{2\bar{L}_{\mathrm{V}}}{\underline{L}_{\mathrm{l}}}(\gamma + (\bar{L}_{\mathrm{V}}L^2 + \bar{L}_{\mathrm{l}} + L_{\mathrm{u}})d_{\mathrm{u}}\sigma_{\mathrm{u1}}^2 + \bar{L}_{\mathrm{V}} d_{\mathrm{x}}\sigma^2 + c_{\mathrm{o}}).$$

**Proof** We conclude from Prop. 3.1 that $\Pr(i_k \neq i^*)$ is bounded by

$$\Pr(i_k \neq i^*) \leq \frac{M^2}{(k-M)^2} \leq \frac{\underline{L}_l}{2\bar{L}_V c_2}, \tag{12}$$

for all $k \geq k_0$. It is important to note that persistence of excitation is only required to hold for $k_0$ steps, as, by our choice of $\eta$, $\Pr(i_k \neq i^*)$ is monotonically decreasing (see proof of Prop. 3.1). By Lemma C.3 we conclude that over the first $k_0$ steps the following holds

$$\mathrm{E}[V(x_{k+1})] \leq c_2 \mathrm{E}[V(x_k)] + \bar{L}_V d_x \sigma^2 + c_o + (\bar{L}_V L^2 + \bar{L}_l) d_u \sigma_{uk}^2,$$

which implies that

$$\mathrm{E}[V(x_k)] \leq c_2^{k_0} k_0 (\bar{L}_V d_x \sigma^2 + c_o + (\bar{L}_V L^2 + \bar{L}_l) d_u \sigma_{u1}^2),$$

for all $k \leq k_0 + 1$, where we have exploited that $\sigma_{uk}$ is monotonically decreasing.

By following the same reasoning (case distinction between $i_k = i^*$ and $i_k \neq i^*$) as in the proof of Thm. 3.2 we arrive at

$$\mathrm{E}[V(x_{k+1})] \leq \mathrm{E}[V(x_k)](\underline{L}_l/(2\bar{L}_V) + 1) + \gamma - \mathrm{E}[l(x_k, u_k)] + \bar{L}_u d_u \sigma_{u1}^2 + \bar{L}_V d_x \sigma^2 + c_o,$$

for all $k \geq k_0$, where we have used inequality (12) to bound $\Pr(i_k \neq i^*)$, $\gamma_k \leq \gamma$, and the fact that $\sigma_{uk}$ is decreasing. The constant $\bar{L}_u$ is given by $\bar{L}_u = \bar{L}_V L^2 + \bar{L}_l + L_u$. Due to the fact that $l$ is bounded below by a quadratic we conclude that $l(x, u) \geq V(x)\underline{L}_l/\bar{L}_V$ for all $x \in \mathbb{R}^{d_x}$, which can be used to simplify the above inequality:

$$\mathrm{E}[V(x_{k+1})] \leq \mathrm{E}[V(x_k)](1 - \underline{L}_l/(2\bar{L}_V)) + \gamma + \bar{L}_u d_u \sigma_{u1}^2 + \bar{L}_V d_x \sigma^2 + c_o.$$

This readily implies

$$\mathrm{E}[V(x_k)] \leq 2\frac{\bar{L}_V}{\underline{L}_l}(\gamma + \bar{L}_u d_u \sigma_{u1}^2 + \bar{L}_V d_x \sigma^2 + c_o),$$

for all $k \geq k_0$, which yields the desired result. $\qquad\square$

## C.5 Convergence in finite time

**Corollary C.6** *(Finite time convergence) Let the assumptions of Prop. 3.1 be satisfied. Then, almost surely, $\{i_k\}_{k=1}^\infty$ converges to $i^*$ in finite time, that is,*

$$\Pr(\sup_{i_k \neq i^*} k < \infty) = 1.$$

**Proof** We conclude from Prop. 3.1 that $\Pr(i_k \neq i^*) \leq M^2/(k-M)^2$ for all $k \geq M + 1$. This implies for any $j \geq M + 1$

$$\Pr(\sup_{i_k \neq i^*} k > j) \leq \sum_{k=j}^\infty \Pr(i_k \neq i^*),$$

where the right-hand side is bounded above by

$$\sum_{k=j}^\infty \frac{M^2}{(k-M)^2} \leq \frac{M^2}{(j-M)^2} + \int_j^\infty \frac{M^2}{(k-M)^2} \mathrm{d}k \leq \frac{M^2}{j-M}\left(1 + \frac{1}{j-M}\right).$$

Hence, the right-hand side converges to zero for large $j$, which yields the desired result. $\qquad\square$

## D Details of Sec. 3.2

The decision-making strategy for S2 is listed in Alg. 2 and proceeds as follows. Alg. 2 first computes a minimizer $\mathrm{argmin}_{\bar{f} \in F} s_k(\bar{f})$, denoted by $f^*$, where $s_k(\bar{f})$ denotes the prediction error as before,

$$s_k(\bar{f}) = \sum_{j=1}^{k-1} \frac{|x_{j+1} - \bar{f}(x_j, u_j)|^2}{1 + |(x_j, u_j)|^2/b^2}, \quad \bar{f} \in F.$$

---

**Algorithm 2** Reinforcement learning (S2)

**Inputs:** $F, \eta, M, \{\sigma_{\mathrm{u}k}^2\}_{k=1}^\infty, \epsilon$
  **for** $k = 1, \ldots$ **do**
    // every $M$ th step
    **if** $\mathrm{mod}(k-1, M) = 0$ **then**
      $f^* \leftarrow \mathrm{argmin}_{\bar{f} \in F}\, s_k(\bar{f})$
      $F^\epsilon \leftarrow \mathrm{greedyCover}(F, f^*, \epsilon)$
      $s_k(f^i) \leftarrow \sum_{j=1}^{k-1} \frac{|x_{j+1} - f^i(x_j, u_j)|^2}{1 + |(x_j, u_j)|^2/b^2}, \, f^i \in F^\epsilon$
      $i_k \sim \exp(-\eta s_k(f^i))/Z, \, f^i \in F^\epsilon$
      compute $\mu^{i_k}$ corr. to $f^{i_k} \in F^\epsilon$ // e.g. by d.p.
    **else**
      $i_k = i_{k-1}$           //stay with $i_{k-1}$
    **end if**
    //follow policy $i_k$ and add excitation
    $u_k = \mu^{i_k}(x_k) + n_{\mathrm{u}k}, \quad n_{\mathrm{u}k} \sim \mathcal{N}(0, \sigma_{\mathrm{u}k}^2 I)$
  **end for**

---

**Algorithm 4** greedyCover($F, f^*, \epsilon$)

$F^\epsilon \leftarrow \{f^*\}$
$S \leftarrow \{\bar{f} \in F \mid \|\bar{f} - f^i\| > \epsilon, \forall f^i \in F^\epsilon\}$

**while** $S \neq \{\}$ **do**
  // pick an element from $S$
  $F^\epsilon \leftarrow F^\epsilon \cup \{\bar{f}\}, \, \bar{f} \in S$
  $S \leftarrow \{\bar{f} \in F \mid \|\bar{f} - f^i\| > \epsilon, \forall f^i \in F^\epsilon\}$
**end while**

**return** $F^\epsilon$

---

In order to apply the same proof arguments as in S1, we will construct an $\epsilon$-cover of $F$ that also includes $f^*$, by running Alg. 4. Alg. 4 greedily adding functions $f^i \in F$ as long as $\|f^i - \bar{f}\| > \epsilon$ for all $\bar{f} \in F_k^\epsilon$, which results in the desired cover. The algorithm then randomly samples $i_k$ as before and applies the feedback policy $\mu^{i_k}$ that corresponds to model $f^{i_k} \in F_k^\epsilon$. Clearly, these steps (minimization over $\bar{f} \in F$, constructing the packing, and solving a dynamic programming problem at every iteration) are computationally intractable in general and one would have to resort to approximations in practice.

We provide regret guarantees next, and therefore slightly modify Ass. 3 from setting S1 as follows.

**Assumption 4** *There exists an integer $M > 0$ and a constant $c_{\mathrm{e}} > 0$ such that for all $x_1 \in \mathbb{R}^{d_{\mathrm{x}}}$, $\sigma_{\mathrm{u}} > 0$, and $f^1, f^2 \in F$,*

$$\frac{1}{M} \sum_{k=1}^M \mathrm{E}\Big[\frac{|f^1(x_k, u_k) - f^2(x_k, u_k)|^2}{1 + |(x_k, u_k)|^2/b^2}\Big] \geq c_{\mathrm{e}} \sigma_{\mathrm{u}}^2 \|f^1 - f^2\|^2,$$

*holds, where $x_{k+1} = f(x_k, u_k) + n_k$, $u_k = \hat{\mu}(x_k) + n_{\mathrm{u}k}$ with $n_k \sim \mathcal{N}(0, \sigma^2 I)$ and $u_k \sim \mathcal{N}(0, \sigma_{\mathrm{u}}^2 I)$, and where $\hat{\mu}$ is any policy corresponding to a model $f \in F$.*

The above assumption differs from Ass. 3 in that the left-hand side involves the discrepancy between a pair of candidate models $f^1$ and $f^2$ instead of that between $f^i$ and the true model $f$. Further, the constant coefficients on the right-hand side also differ. The assumption is stated for a finite $b > 0$ even though it can be relaxed to $b \to \infty$, and the same policy-regret guarantees apply although with more elaborate constants, see App. F for further discussion. For $b \to \infty$ the assumption describes persistence of excitation as used in system identification and statistics (see, e.g., Ljung, 1999, Ch. 8.2). From a maximum-likelihood point of view, Ass. 8 ensures that the dynamics $f$ correspond to a unique non-degenerate minimum of the one-step prediction error, accumulated over $M$ steps. The assumption is generically satisfied if the models $f \in F$ are linear and $\|\cdot\|$ denotes the Lipschitz-norm (see, e.g., Beer & Hoffman, 2013), as highlighted in Sec. 3. A similar reasoning applies to nonlinear systems, see Sec. F.

We will further strengthen the Bellman-inequality from Ass. 1 to ensure that the steady-state performance $\gamma$ of the policy $\mu$ is stable under small policy changes that arise from models $f^i \in F$ close to $f$. This notion of stability requires $\mu$ to optimize the corresponding Q-function. This is made precise as follows.

**Assumption 5** *(Bellman-type inequality) For all small enough $\xi > 0$ there exists a cost-to-go function $V$ (corresponding to $f$ and $\mu$) satisfying the following inequality:*

$$V(x) \geq \mathrm{E}[l(x, \mu^i(x) + n_{\mathrm{u}}) + V(f(x, \mu^i(x) + n_{\mathrm{u}}) + n))] - \gamma - L_{\mathrm{u}} d_{\mathrm{u}} \sigma_{\mathrm{u}}^2 - L_\mu \xi^2,$$

*for all policies $\mu^i$ corresponding to $\|f^i - f\| < \xi$, for all $x \in \mathbb{R}^{d_{\mathrm{x}}}$, where $L_{\mathrm{u}}, L_\mu > 0$ are constant, $n \sim \mathcal{N}(0, \sigma^2 I)$, and $n_{\mathrm{u}} \sim \mathcal{N}(0, \sigma_{\mathrm{u}}^2 I)$.*

The following proposition provides a sufficient condition for Ass. 5 to hold. In particular, the proposition applies to the class of linear dynamical systems with a quadratic, positive definite stage cost, where all assumptions are satisfied (see also Prop. 6 in Simchowitz & Foster, 2020).

**Proposition D.1** *Let Ass. 1 and Ass. 2 be satisfied and fix $x \in \mathbb{R}^{d_x}$. If, in addition,*

$$l(x,u) \geq \underline{L}_l |x|^2/2, \quad \mu(x) \in \underset{u \in \mathbb{R}^{d_u}}{\arg\min} \, \mathrm{E}[l(x,u) + V(f(x,u)+n)], \quad and \quad \|\mu^i - \mu\|_{\mathrm{op}} \leq L'_\mu \xi,$$

*holds for all policies $\mu^i$ corresponding to $\|f^i - f\|_{\mathrm{op}} < \xi$ and all $\xi > 0$ small enough, then Ass. 5 is satisfied for $x$ and all $\sigma_u$ small enough, where $\underline{L}_l, L'_\mu > 0$ are constant and $n \sim \mathcal{N}(0, \sigma^2 I)$. The Lipschitz-norm $\| \cdot \|_{\mathrm{op}}$ is defined for any Lipschitz-continuous function $q : \mathbb{R}^{d_x} \to \mathbb{R}^d$ as*

$$\|q\|_{\mathrm{op}} := \max \left\{ |q(0)|, \sup_{x_1, x_2 \in \mathbb{R}^{d_x}} \frac{|q(x_1) - q(x_2)|}{|x_1 - x_2|} \right\}.$$

**Proof** Let $c_s/L'^2_\mu$ denote the smoothness constant of $\mathrm{E}[l(x,u) + V(f(x,u)+n)]$ in $u$ in a neighborhood of $u = \mu(x)$. From smoothness and the fact that $\mu(x)$ is a minimizer we conclude

$$\mathrm{E}[l(x,\mu^i(x)) + V(f(x,\mu^i(x))+n)] \leq \mathrm{E}[l(x,\mu(x)) + V(f(x,\mu(x))+n)] + c_s |\mu^i(x) - \mu(x)|^2/(2L'_\mu)$$

$$\leq V(x) + \gamma + c_s |\mu^i(x) - \mu(x)|^2/(2L'^2_\mu), \tag{13}$$

where Ass. 1 has been used for the second step (where we set $\sigma_u = 0$). We further note that

$$|\mu^i(x) - \mu(x)| = |\mu^i(0) - \mu(0) + \mu^i(x) - \mu(x) - (\mu^i(0) - \mu(0))| \leq L'_\mu(\xi + \xi|x|),$$

due to the fact that $\|\mu^i - \mu\|_{\mathrm{op}} \leq L'_\mu \xi$. We multiply (13) by $1 + 4c_s \xi^2/\underline{L}_l$, define $\tilde{V} := (1 + 4c_s\xi^2/\underline{L}_l) V$, and arrive at

$$\mathrm{E}[l(x,\mu^i(x)) + \tilde{V}(f(x,\mu^i(x))+n)] \leq \tilde{V}(x) + (1 + 4c_s\xi^2/\underline{L}_l)\gamma - 2\xi^2 c_s |x|^2$$
$$+ c_s(1 + 4c_s\xi^2/\underline{L}_l)\xi^2 + c_s(1 + 4c_s\xi^2/\underline{L}_l)\xi^2|x|^2,$$

where we have used the fact that $l(x,u) \geq \underline{L}_l|x|^2/2$. We choose $\xi^2 \leq \underline{L}_l/(4c_s)$ and rearrange terms. This results in

$$\mathrm{E}[l(x,\mu^i(x)) + \tilde{V}(f(x,\mu^i(x))+n)] \leq \tilde{V}(x) + \gamma + 2c_s(2\gamma/\underline{L}_l + 1)\xi^2 + c_s(-1 + 4c_s\xi^2/\underline{L}_l)\xi^2|x|^2,$$

where the last term is non-positive. We therefore conclude that the inequality in Ass. 5 holds for $\tilde{V}$ with $L_\mu = 2c_s(2\gamma/\underline{L}_l + 1)$, $L_u = c_s/(2L'^2_\mu)$, and a small enough $\sigma_u$. $\qquad \square$

We are now ready to prove the main result of this section:

**Theorem D.2** *Let Ass. 2, Ass. 4, and Ass. 5 be satisfied and choose $\eta$ and $\sigma_{uk}$ as*

$$\eta = \min \left\{ \frac{1}{4M\sigma^2}, \frac{1}{2ML^2b^2} \right\}, \quad \sigma^2_{uk} = \frac{4}{\eta c_e M \epsilon^2} \left( \frac{2}{\lceil k/M \rceil} + \frac{\ln(m(\epsilon))}{(\lceil k/M \rceil)^2} \right).$$

*Then, the policy regret of Alg. 2 is bounded by*

$$\sum_{k=1}^{N} \mathrm{E}[l(x_k, u_k)] - N\gamma \leq c_{r1} d_u \frac{3\ln(N) + M\ln(m(\epsilon))}{\epsilon^2} + L_\mu N \epsilon^2 + c_{r2}$$

*for all $N \geq 2M$, where $m(\epsilon)$ denotes the packing number of $F$ for a packing of size $\epsilon$ and where the constants $c_{r1}$ and $c_{r2}$ are given by*

$$c_{r1} = \frac{8c_\alpha(\bar{L}_V L^2 + \bar{L}_l + L_u)}{\eta c_e}, \quad c_\alpha = e^{3c_2 M}, \quad c_{r2} = 3Mc_\alpha(\bar{L}_V d_x \sigma^2/2 + c_o).$$

**Proof** The proof follows Thm. 3.2. At every iteration $k$ we denote by $I^*_k$ the set of models $f^{i^*} \in F^\epsilon_k$ that satisfy $\|f^{i^*} - f\| \leq \epsilon$. We then conclude from the same reasoning as in Prop. 3.1 that

$$\Pr(i_k \notin I^*_k) \leq \frac{M^2}{(k-M)^2},$$

for all $k \geq M + 1$. We make therefore the case distinction $i_k \in I_k^*$ and $i_k \notin I_k^*$, which then yields by the same arguments (see (9))

$$\sum_{k=1}^{N}(\mathrm{E}[l(x_k, u_k)] - \gamma_k) \leq NL_\mu\epsilon^2 + c_\alpha \bar{L}_{\mathrm{u}} d_{\mathrm{u}} \sum_{k=1}^{N}\sigma_{\mathrm{u}k}^2 + c_\alpha(\bar{L}_{\mathrm{V}} d_{\mathrm{x}}\sigma^2/2 + c_{\mathrm{o}}) \sum_{k=1}^{N}\mathrm{Pr}(i_k \notin I_k^*),$$

where there is an additional error term, due to the fact that $f^{i^*}$ and $f$ could be different (although $\|f^{i^*} - f\| \leq \epsilon$ for any $i^* \in I_k^*$, by construction of $I_k^*$). The desired result follows from the previous inequality. However, compared to Thm. 3.2 we used the slightly more conservative bound

$$\sum_{k=1}^{N}\sigma_{\mathrm{u}k}^2 \leq \frac{8}{\eta c_{\mathrm{e}}\epsilon^2}(3\ln(N) + M\ln(m(\epsilon))),$$

which applies as long as $N \geq 2$, and simplifies the resulting constants. $\qquad\square$

# E  DETAILS OF SEC. 3.3

For deriving the regret bound we will slightly adapt the persistence of excitation condition Ass. 3 from Sec. 3. The motivation is analogous to Ass. 4 and we refer the reader to App. D and App. F for further discussion.

**Assumption 6** *There exists an integer $M > 0$ and a constant $\bar{c}_{\mathrm{e}} > 0$ such that*

$$\frac{1}{M}\sum_{k=1}^{M}\mathrm{E}\Big[\frac{|f_\theta(x_k, u_k) - f(x_k, u_k)|^2}{1 + |(x_k, u_k)|^2/b^2}\Big] \geq \bar{c}_{\mathrm{e}}\sigma_{\mathrm{u}}^2|\theta|^2,$$

*for all $\theta \in \Omega$ and all $x_1 \in \mathbb{R}^{d_{\mathrm{x}}}$, where $x_{k+1} = f(x_k, u_k) + n_k$, $u_k = \mu_\theta(x_k) + n_{\mathrm{u}k}$, and $n_k, n_{\mathrm{u}k}$ are independent random variables that satisfy $n_{\mathrm{u}k} \sim \mathcal{N}(0, \sigma_{\mathrm{u}}^2 I)$, $n_k \sim \mathcal{N}(0, \sigma^2 I)$.*

Intuitively, the above uniform lower-boundedness property holds if the difference between candidate parametric models is of the same order as the difference between their parameters (by following a similar derivation as Prop. F.2).

The second assumption, which will be important, is a strengthened version of the Bellman-inequality from Ass. 1. The assumption ensures that the steady-state performance $\gamma$ of the policy $\mu$ is stable under small policy changes that arise from models $f_\theta \in F$ that are close to $f$. The sufficient condition provided by Prop. D.1 applies here in the same way ($\|f_\theta - f\|_{\mathrm{op}}$ reduces to $|\theta|$) and we therefore conclude that the assumption below is, for example, satisfied for linear dynamical systems with a quadratic, positive definite stage cost.

**Assumption 7** *(Bellman-type inequality) For all small enough $\xi > 0$, there exists a cost to go function $V$ (corresponding to $f$ and $\mu$) satisfying the following inequality:*

$$V(x) \geq \mathrm{E}[l(x, \mu_\theta(x) + n_{\mathrm{u}}) + V(f(x, \mu_\theta(x) + n_{\mathrm{u}}) + n)] - \gamma - L_{\mathrm{u}} d_{\mathrm{u}}\sigma_{\mathrm{u}}^2 - L_\mu\xi^2,$$

*for all policies $\mu_\theta$ with $|\theta| < \xi$, for all $x \in \mathbb{R}^{d_{\mathrm{x}}}$, where $L_{\mathrm{u}}, L_\mu > 0$ are constant, $n \sim \mathcal{N}(0, \sigma^2 I)$, and $n_{\mathrm{u}} \sim \mathcal{N}(0, \sigma_{\mathrm{u}}^2 I)$.*

We now prove the main result characterizing policy regret for the setting S3.

**Theorem E.1** *Let Ass. 2, Ass. 6, and Ass. 7 be satisfied and choose $\eta$ and $\sigma_{\mathrm{u}k}^2$ as*

$$\eta \leq \min\Big\{\frac{1}{4M\sigma^2}, \frac{1}{2ML^2 b^2}\Big\}, \quad \sigma_{\mathrm{u}k}^2 = \frac{4}{\eta\bar{c}_{\mathrm{e}} M\epsilon^2}\Big(\frac{2}{\lceil k/M\rceil} + \frac{p}{(\lceil k/M\rceil)^2}\Big).$$

*Then, for all $N \geq 2M$ there exists a large enough $p$, such that the policy regret of Alg. 3 is bounded by*

$$\sum_{k=1}^{N}\mathrm{E}[l(x_k, u_k)] - N\gamma \leq 2\sqrt{c_{\mathrm{r}1}(3\ln(N) + Mp)d_{\mathrm{u}}N} + c_{\mathrm{r}2},$$

*where the constants $c_{r1}$ and $c_{r2}$ are given by*

$$c_{r1} = \frac{8c_\alpha L_\mu(\bar{L}_V L^2 + \bar{L}_l + L_u)}{\eta \bar{c}_e}, \quad c_\alpha = e^{3c_2 M}, \quad c_{r2} = 3Mc_\alpha(\bar{L}_V d_x \sigma^2/2 + c_o),$$

*with*

$$\epsilon^2 = \sqrt{\frac{c_{r1} d_u(3\ln(N) + Mp)}{L_\mu^2 N}}.$$

**Proof** We first argue that the reasoning in Lemma C.1 applies in a very similar way to setting $\mathsf{S}3$. To that extent, we first define the random variable $p_k$ as follows

$$p_k = \frac{\int_{\Omega \backslash \{\theta: |\theta| \leq \epsilon\}} e^{-\eta(s_k(\theta) - \bar{s}_k)} d\theta}{\int_\Omega e^{-\eta(s_k(\theta) - \bar{s}_k)} d\theta},$$

where $d\theta$ denotes the Lebesgue measure, and $\bar{s}_k = \min_{\theta \in \Omega} s_k(\theta)$. However, compared to the discrete setting, where the denumerator was simply bounded below by unity, the situation is more delicate. More precisely, we bound the denumerator from below by $|B_\delta| e^{-\eta h(\delta)}$, where $h(\delta) := \max_{\theta \in B_\delta} s_k(\theta) - \bar{s}_k$ and $B_\delta$ denotes a ball of radius $\delta$ with volume $|B_\delta|$ centered at a minimizer of $s_k(\theta)$. From the smoothness of $s_k(\theta)$ we conclude that $h(\delta) = \mathcal{O}(\delta^2)$ for small $\delta$. Due to our normalization, $\Omega$ is contained in a ball of unit radius and we have $|B_\delta|/|\Omega| \geq |B_\delta|/|B_1| \geq \delta^p$ where $p$ refers to the dimension of $\Omega$. Hence we arrive at the following lower bound

$$\int_\Omega e^{-\eta(s_k(\theta) - \bar{s}_k)} d\theta \geq |\Omega| \delta^p e^{-h(\delta)} \gtrsim |\Omega| e^{-p},$$

where the second inequality arises from carefully choosing $\delta$ in order to balance the the term $\delta^p$ and $e^{-h(\delta)}$. This yields the following bound on $p_k$ (which resembles the discrete setting)

$$p_k \leq \frac{e^p}{|\Omega|} \int_{\Omega \backslash \{\theta: |\theta| \leq \epsilon\}} e^{-\eta(s_k(\theta) - s_k^*)} d\theta,$$

where we have also replaced $\bar{s}_k$ with $s_k^*$ due to the fact that $\bar{s}_k$ is a minimum. Following the same reasoning as in Lemma C.1 and Prop. 3.1 yields therefore

$$\Pr(|\theta_k| > \epsilon) \leq \frac{e^p}{|\Omega|} \int_{\Omega \backslash \{\theta: |\theta| \leq \epsilon\}} \mathrm{E}[e^{-\eta(s_k(\theta) - s_k^*)}] d\theta \leq e^p \exp\left(-\frac{\bar{c}_e \eta}{4} \epsilon^2 \sum_{j=1}^{k-M} \sigma_{uj}^2\right),$$

where Fubini's theorem has been used in the first step to interchange expectation and integration. In addition, due to the modification of $\sigma_{uk}^2$ compared to Thm. D.2 (where now $m(\epsilon)$ is replaced by $e^p$), we find that

$$\Pr(|\theta_k| \geq \epsilon) \leq \frac{M^2}{(k-M)^2},$$

for all $k \geq M + 1$. We apply the same reasoning as in the proof of Thm. 3.2, where we now have the case distinction $\Pr(|\theta_k| > \epsilon)$ and $\Pr(|\theta_k| \leq \epsilon)$ (corresponding to $\Pr(i_k \neq i^*)$ and $\Pr(i_k = i^*)$). This concludes that

$$\sum_{k=1}^N (\mathrm{E}[l(x_k, u_k)] - \gamma_k) \leq L_\mu N \epsilon^2 + c_\alpha \bar{L}_u d_u \sum_{k=1}^N \sigma_{uk}^2 + c_\alpha(\bar{L}_V d_x \sigma^2/2 + c_o) \sum_{k=1}^N \Pr(|\theta_k| > \epsilon),$$

where, as before,

$$\sum_{k=1}^N \Pr(|\theta_k| > \epsilon) \leq 3M.$$

We further note that the sum over $\sigma_{uk}^2$ yields

$$\sum_{k=1}^N \sigma_{uk}^2 \leq \frac{8}{\eta \bar{c}_e \epsilon^2}(1 + \ln(N) + Mp) \leq \frac{8}{\eta \bar{c}_e \epsilon^2}(3\ln(N) + Mp),$$

where $N \geq 2M \geq 2$ (and therefore $\ln(N) \geq 1/2$) has been used in the second step. Consequently the regret is bounded by

$$\sum_{k=1}^{N} \mathrm{E}[l(x_k, u_k)] - N\gamma \leq L_\mu N\epsilon^2 + c_{\mathrm{r}1} d_{\mathrm{u}} \frac{3\ln(N) + Mp}{L_\mu \epsilon^2} + c_{\mathrm{r}2}.$$

The choice of $\epsilon$ achieves an optimal trade-off between the first two terms, which yields the desired result. $\qquad\square$

## F    RELAXING PERSISTENCE OF EXCITATION

This section discusses the situation when $b \to \infty$. We first note that when $f^i \in F$ are linear (see also Chatzikiriakos et al., 2025), that is $f^i(x, u) = A^i x + B^i u$, $\mu^i(x) = K^i x$, Ass. 3 for $b \to \infty$ is straightforward to verify and we obtain, for example, the following bound for $k \geq 2$:

$$\mathrm{E}[|f^i(x_k, u_k) - f(x_k, u_k)|^2] \geq \sigma_{\mathrm{u}}^2 |B^i - B|_{\mathrm{F}}^2 + \left(\sigma_{\mathrm{u}}^2 \underline{\sigma}(W_{k-1}^{\mathrm{c}}) + \sigma^2\right) |A^i - A + (B^i - B)K^q|_{\mathrm{F}}^2, \tag{14}$$

where $K^q$ represents the linear feedback controller corresponding to model $f^q \in F$, and $W_k^{\mathrm{c}}$ denotes the controllability Gramian (over $k$ steps):

$$W_k^c = \sum_{j=0}^{k-1} (A + BK^q)^{j^\top} BB^\top (A + BK^q)^j,$$

where $\underline{\sigma}$ denotes the minimum singular value. Hence, we conclude that Ass. 3 is satisfied for $M = 2$ with $\Delta = \min_{i,q \in \{1,\ldots,m\}} |B^i - B|_{\mathrm{F}}^2 + \underline{\sigma}(W_{k-1}^{\mathrm{c}})|A^i - A + (B^i - B)K^q|_{\mathrm{F}}^2$.

We now derive a variant of Lemma C.2 that relies on the fact that over finite time $x_k$ and $u_k$ are sub-Gaussian random variables. To this extent, we slightly modify Ass. 3 as follows:

**Assumption 8** *There exists an integer $M > 0$ and a constant $\Delta > 0$ such that for any $x_1 \in \mathbb{R}^{d_{\mathrm{x}}}$, $\sigma_{\mathrm{u}} > 0$, and $f^i \in F$, $f^i \neq f$,*

$$\frac{1}{M} \sum_{k=1}^{M} \mathrm{E}\left[|f^i(x_k, u_k) - f(x_k, u_k)|^2\right] \geq \Delta(\sigma_{\mathrm{u}}^2 + d_{\mathrm{x}}\sigma^2)$$

*holds, where $x_{k+1} = f(x_k, u_k) + n_k$, $u_k = \mu^q(x_k) + n_{\mathrm{u}k}$ with $n_k \sim \mathcal{N}(0, \sigma^2 I)$, $n_{\mathrm{u}k} \sim \mathcal{N}(0, \sigma_{\mathrm{u}}^2 I)$, and $q \in \{1, \ldots, m\}$.*

**Lemma F.1** *Let Ass. 8 be satisfied and let*

$$l_k^i = |x_{k+1} - f^i(x_k, u_k)|^2,$$

*for $k = 1, 2, \ldots$, where $x_k, u_k$ denotes the trajectory resulting from Alg. 1, and $\sigma_{\mathrm{u}k}^2$ is monotonically decreasing. Let $k' \geq 0$ be an integer and define $k = k'M + 1$ (i.e., $k$ is a time instance where $i_k$ switches). Then, the following bound holds for all $0 < \eta \leq \min\{1/(4M\sigma^2), \eta_0\}$ and all $j = k, \ldots, k + M - 1$*

$$\mathrm{E}[e^{-\eta \sum_{j=k}^{k+M-1}(l_j^i - l_j^*)}|x_k] \leq \exp\left(-\frac{\eta\Delta}{4} \sum_{j=k}^{k+M-1} (\sigma_{\mathrm{u}j}^2 + d_{\mathrm{x}}\sigma^2/d_{\mathrm{u}})\right),$$

*where*

$$\eta_0 = \frac{1}{4M(\sigma^2 d_{\mathrm{x}} + \sigma_{\mathrm{u}1}^2 d_{\mathrm{u}})} \cdot \frac{\Delta}{128M^2 d_{\mathrm{u}}(2L^{2M}(1 + L_\mu)^{2M})^2},$$

*and $f_j, f_j^i$ is shorthand notation for $f(x_j, u_j)$ and $f^i(x_j, u_j)$, respectively.*

**Proof** Without loss of generality we set $k = 1$ and $k' = 0$ (the proof follows exactly the same steps for $k' > 0$). We first note that the random variables $x_j$ and $u_j$ for $j \geq k$ are Lipschitz

continuous functions of the noise variables $\{n_q, n_{\mathrm{u}q}\}_{q=1}^{M-1}$. We define the random variable $X_j := |f(x_j, u_j) - f^i(x_j, u_j)|/\sqrt{2}$ and note that $X_j$ is sub-Gaussian with variance proxy

$$\tilde{\sigma}^2 := 2L^{2M}(1 + L_\mu)^{2M} M(\sigma_{\mathrm{u}1}^2 d_{\mathrm{u}} + \sigma^2 d_{\mathrm{x}}),$$

due to the fact that there are at most $M$ steps between $x_1$ and $x_j$. We will simplify the notation by introducing the following variables

$$\tilde{L} := 2L^{2M}(1 + L_\mu)^{2M}, \sigma_{\mathrm{e}}^2 := M(\sigma_{\mathrm{u}1}^2 d_{\mathrm{u}} + \sigma^2 d_{\mathrm{x}}),$$

such that $\tilde{\sigma}^2 = \tilde{L}\sigma_{\mathrm{e}}^2$ and $128M^2\tilde{\sigma}^2\eta_0 = \Delta/(4\tilde{L}d_{\mathrm{u}})$. The previous result exploits the fact that a $L_{\mathrm{v}}$-Lipschitz function of a set of $p_{\mathrm{v}}$ independent standard Gaussian random variables is sub-Gaussian with variance proxy $p_{\mathrm{v}}L_{\mathrm{v}}$ (see, e.g., Wainwright, 2019, Ch.2.3). By following the same argument as in Prop. 3.1 and Lemma. C.2 we arrive at

$$\mathrm{E}[e^{-\eta\sum_{j=1}^M(l_j^i - l_j^*)}] \leq \left(\prod_{j=1}^M \mathrm{E}[e^{-\eta M(l_j - l_j^*)}]\right)^{1/M}$$

$$\leq \left(\prod_{j=1}^M \mathrm{E}[e^{-\eta M|f_j - f_j^i|^2/2}]\right)^{1/M}$$

$$\leq \left(\prod_{j=1}^M \mathrm{E}[e^{-\eta M X_j^2}]\right)^{1/M},$$

where we used the shorthand notation $f_j, f_j^i$ as in the statement of the lemma and the fact $\eta \leq 1/(4M\sigma^2)$. The random variables $X_j$ are sub-Gaussian with variance proxy $\tilde{\sigma}^2$ and therefore $X_j^2 - \mathrm{E}[X_j^2]$ are sub-Exponential with parameter $16\tilde{\sigma}^2$. As a result, we can simplify the previous inequality to

$$\mathrm{E}[e^{-\eta\sum_{j=1}^M(l_j^i - l_j^*)}] \leq \left(\prod_{j=1}^M e^{-\eta M \mathrm{E}[X_j^2] + \eta\tilde{\sigma}^2\Delta/(4\tilde{L}d_{\mathrm{u}})}\right)^{1/M},$$

since $128M\eta\tilde{\sigma}^2 \leq \Delta/(4\tilde{L}d_{\mathrm{u}})$ by our choice of $\eta_0$. As a result of Ass. 8 we infer

$$\sum_{j=1}^M \mathrm{E}[X_j^2] \geq M\Delta(\sigma_{\mathrm{u}1}^2 + d_{\mathrm{x}}\sigma^2)/2 \geq \Delta\sigma_{\mathrm{e}}^2/(2d_{\mathrm{u}}),$$

and therefore

$$\sum_{j=1}^M \mathrm{E}[X_j^2] - \frac{\Delta\tilde{\sigma}^2}{4\tilde{L}d_{\mathrm{u}}} \geq \sigma_{\mathrm{e}}^2\left(\frac{\Delta}{2d_{\mathrm{u}}} - \frac{\Delta}{4d_{\mathrm{u}}}\right) = \frac{\sigma_{\mathrm{e}}^2\Delta}{4d_{\mathrm{u}}}.$$

This establishes

$$\mathrm{E}[e^{-\eta\sum_{j=1}^M(l_j^i - l_j^*)}] \leq e^{-\eta\Delta\sigma_{\mathrm{e}}^2/(4d_{\mathrm{u}})},$$

and yields the desired result. $\qquad\square$

The conclusion from the setting with linear dynamics can be generalized to nonlinear systems as follows. In order to simplify the presentation we consider the situation where the process noise is absent ($\sigma = 0$); the same rationale applies when $\sigma > 0$. The following result demonstrates that Ass. 3 is generic and holds for a broad class of nonlinear dynamics. This result also highlights a close connection between controllability and the required notion of persistence of excitation.

**Proposition F.2** *Let $x \in \mathbb{R}^{d_{\mathrm{x}}}$ and $q \in \{1, \ldots, m\}$ be fixed. Then, there exists a constant $\Delta' > 0$ such that*

$$\mathrm{E}[|f^i(x_k, u_k) - f(x_k, u_k)|^2] \geq \Delta'\sigma_{\mathrm{u}}^2$$

*for all small enough $\sigma_{\mathrm{u}} > 0$ if either of the two inequalities are satisfied*

$$|f^i(\bar{x}_k, \mu^q(\bar{x}_k)) - f(\bar{x}_k, \mu^q(\bar{x}_k))|^2 > 0, \qquad \underline{\sigma}(W_{k-1}^{\mathrm{c}})|A_k^i - A_k|_{\mathrm{F}}^2 + |B_k^i - B_k|_{\mathrm{F}}^2 > 0,$$

*where $x_k$ is defined recursively via $x_1 = x$, $x_{j+1} = f(x_j, \mu^q(x_j) + n_{uj})$ with $n_{uj} \sim \mathcal{N}(0, \sigma^2_{uj} I)$ $j = 1, \ldots, k-1$ and*

$$A_k := \frac{\partial}{\partial x} f(x, \mu^q(x)) \Big|_{x=\bar{x}_k}, \quad A_k^i := \frac{\partial}{\partial x} f^i(x, \mu^q(x)) \Big|_{x=\bar{x}_k},$$

$$B_k := \frac{\partial}{\partial u} f(x, u) \Big|_{x=\bar{x}_k, u=\mu^q(\bar{x}_k)}, \quad B_k^i := \frac{\partial}{\partial u} f(x, u) \Big|_{x=\bar{x}_k, u=\mu^q(\bar{x}_k)},$$

$$W_k^c := \sum_{j=1}^{k-1} A_{k-1} A_{k-2} \ldots A_{j+1} B_j B_j^\top A_{j+1}^\top \ldots A_{k-2}^\top A_{k-1}^\top.$$

*Moreover, $\bar{x}_k$ corresponds to the noise-free trajectory and is defined via $\bar{x}_1 = x, \bar{x}_{j+1} = f(\bar{x}_j, \mu^q(\bar{x}_j))$.*

**Proof** We start by considering the situation where $f^i(\bar{x}_k, \mu^q(\bar{x}_k)) \neq f(\bar{x}_k, \mu^q(\bar{x}_k))$. We note that

$$\mathrm{E}[|f^i(x_k, \mu^q(x_k) + n_{uk}) - f(x_k, \mu^q(x_k) + n_{uk})|^2]$$

continuously depends on $\sigma_u$ and converges to $|f^i(\bar{x}_k, \mu^q(\bar{x}_k)) - f(\bar{x}_k, \mu^q(\bar{x}_k))|^2 > 0$ as $\sigma_u \to 0$. Hence the desired inequality is clearly satisfied for all small enough $\sigma_u > 0$.

Next we consider the situation where $f^i(\bar{x}_k, \mu^q(\bar{x}_k)) = f(\bar{x}_k, \mu^q(\bar{x}_k))$ and apply Taylor's theorem as follows:

$$f(x_k, \mu^q(x_k) + n_{uk}) = f(\bar{x}_k, \mu^q(\bar{x}_k)) + A_k(\bar{x}_k - x_k) + B_k n_{uk} + o(\bar{x}_k - x_k, n_{uk}),$$

where $o$ is a continuous function that satisfies $o(\xi)/|\xi| \to 0$ for $|\xi| \to 0$. We therefore conclude

$$|f^i(x_k, u_k) - f(x_k, u_k)| = \left| (A_k^i - A_k)(x_k - \bar{x}_k) + (B_k^i - B_k) n_{uk} + o(\bar{x}_k - x_k, n_{uk}) \right|,$$

where we slightly abused notation to redefine the reminder term (we will frequently do so throughout the remainder of the proof). We further apply Taylor's theorem to express $x_k - \bar{x}_k$ as

$$x_k - \bar{x}_k = \sum_{j=1}^{k-1} A_{k-1} \ldots A_{j+1} B_j n_{uj} + o(n_{u1}, \ldots, n_{uk-1}).$$

By combining the previous two equations, squaring, and taking expectations, we arrive at

$$\mathrm{E}[|f^i(x_k, u_k) - f(x_k, u_k)|^2] \geq (|A_k^i - A_k|_F^2 \underline{\sigma}(W_{k-1}^c) + |B_k^i - B_k|_F^2) \sigma_u^2 - o(\sigma_u^2),$$

where we took advantage of the fact that $n_{u1}, \ldots, n_{uk}$ are mutually independent. We further used the following reasoning: i) independence between $n_{ui}$ and $n_{uj}$, $i \neq j$, concludes

$$\mathrm{E}[o(n_{ui}) n_{uj}^\top] = \mathrm{E}[o(n_{ui}) \mathrm{E}[n_{uj}^\top \mid n_{ui}]] = 0.$$

ii) for $i = j$ we have

$$\mathrm{E}[o(|n_{ui}|^2)] = \int_{\mathbb{R}^{d_u}} o(|\xi|^2) \frac{1}{(\sqrt{2\pi}\sigma_u)^{d_u}} e^{-|\xi|^2/(2\sigma_u^2)} \mathrm{d}\xi \leq \underbrace{\int_{\mathbb{R}^{d_u}} o(|\xi|^2) \frac{2^q}{\sqrt{2\pi}^{d_u}} \frac{q!}{|\xi|^{2q}} \mathrm{d}\xi}_{=\text{const.}} \sigma_u^{2q-d_u},$$

for any integer $q \geq 0$ large enough, where we have bounded the exponential using $e^{-\xi} \leq q!/\xi^q$ for all $\xi \geq 0$. This implies that $\mathrm{E}[o(|n_{ui}|^2)] = o(\sigma_u^2)$ (in fact $\mathrm{E}[o(|n_{ui}|^2)]$ decays much faster for small $\sigma_u$), which leads to the desired result. $\square$

## G  DISCUSSION OF REALIZABILITY ASSUMPTION

The realizability assumption, i.e., $f \in F$, can be relaxed in multiple ways and is an active area of research in statistical learning theory. We refer the reader to van Erven et al. (2015) for a detailed and thorough discussion. In the following we provide two simple conditions for relaxing the realizability condition. We begin with the Setting S1, followed by Setting S2 and S3.

## G.1 Setting S1

**Assumption** (Setting S1) There exists a constant $c_{\mathrm{r}} > 0$ and a candidate model $f^* \in \{f^1, \ldots, f^m\}$ such that
$$(1 + 2c_{\mathrm{r}})|f^*(x, u) - f(x, u)| \leq |\bar{f}(x, u) - f(x, u)|,$$
for all $(x, u) \in \mathbb{R}^{d_{\mathrm{x}} + d_{\mathrm{u}}}$ and for all $\bar{f} \in \{f^1, \ldots, f^m\} \setminus \{f^*\}$.

The assumption implies that there is a single candidate model that is more accurate than the others over the entire state-action space by a fixed margin $2c_{\mathrm{r}}$. The graphical intuition of the assumption is shown in Fig. 6. It is easy to see that the assumption implies[5]

$$2(f^*(x, u) - f(x, u))^\top (f^i(x, u) - f^*(x, u)) \geq -\frac{1}{1 + c_{\mathrm{r}}}|f^*(x, u) - f^i(x, u)|^2. \tag{15}$$

As a consequence, the proof of Lemma C.2 remains valid with minor modifications. More precisely, we obtain

$$l_k^i - l_k^* = \frac{2(f^* - f)^\top (f^i - f^*) + |f^* - f^i|^2 + 2(f^* - f^i)^\top n_k}{1 + |(x_k, u_k)|^2/b^2}$$
$$\geq \frac{\frac{c_{\mathrm{r}}}{1 + c_{\mathrm{r}}}|f^* - f^i|^2 + 2(f^* - f^i)^\top n_k}{1 + |(x_k, u_k)|^2/b^2},$$

where the dependency of $f$, $f^i$, and $f^*$ on $(x_k, u_k)$ has been omitted. The remaining part of the proof of Lemma C.2 applies in the same way with minor changes in the parameter $\eta$. This also implies that the remaining results, i.e., Thm. 2.1 and Thm. 3.2 follow in an analogous manner, with a benchmark given by the cumulative cost of any policy based on the above closest model $f^*$.

## G.2 Setting S2 and S3

For the Setting S2 and Setting S3 the realizability assumption can be relaxed in the following way:

**Assumption** (Setting S2 and S3) There exists a candidate model $f^* \in F$ such that
$$(f^*(x, u) - f(x, u))^\top (\bar{f}(x, u) - f^*(x, u)) \geq 0,$$
holds for all $(x, u) \in \mathbb{R}^{d_{\mathrm{x}} + d_{\mathrm{u}}}$ and for all $\bar{f} \in F$.

The geometric interpretation of the assumption is shown in Fig. 6, and the assumption is again slightly stronger than necessary (see (15)). As a result of the assumption the proof of Lemma C.2 applies in exactly the same way as

$$l_k^i - l_k^* = \frac{2(f^* - f)^\top (f^i - f^*) + |f^* - f^i|^2 + 2(f^* - f^i)^\top n_k}{1 + |(x_k, u_k)|^2/b^2}$$
$$\geq \frac{|f^* - f^i|^2 + 2(f^* - f^i)^\top n_k}{1 + |(x_k, u_k)|^2/b^2},$$

where the dependency of $f$, $f^i$, and $f^*$ on $(x_k, u_k)$ has been omitted. Hence, the remaining results, i.e., Thm. 2.2 and Thm. 2.3 apply in exactly the same way, where the benchmark therein corresponds to the cumulative cost incurred by any policy associated with $f^*$.

---

[5]The converse is not true, so the assumption is slightly stronger than strictly necessary. The condition (15) has a clear geometric interpretation, which is, however, slightly more involved.

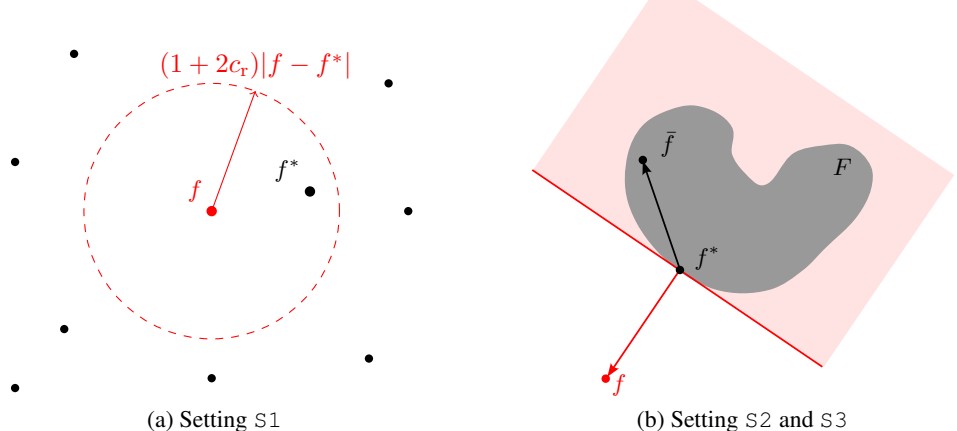

(a) Setting S1
(b) Setting S2 and S3

Figure 6: The figure provides a graphical illustration on the two assumptions in App. G.1 and App. G.2. In the Setting S2 and S3 the set $F$ is assumed to be connected, however, this does not need to be the case. Convexity of $F$ is a sufficient condition for the assumption in App. G.2.

