# OpenReview forum: "The Sample Complexity of Online Reinforcement Learning: A Multi-model Perspective"
_ICLR.cc/2026/Conference — ICLR 2026 Poster_

### Official Review · Reviewer_Zps8 · 2025-10-28

**Soundness:** 4
**Presentation:** 3
**Contribution:** 3
**Rating:** 6
**Confidence:** 4

**Summary:**

This paper presents a theoretical study of the problem of online reinforcement learning in general (non-linear) continuous systems from the perspective of sample complexity. For context, this version of RL is the non-episodic setting which arose from generalising multi-armed bandits: one competes via regret with the best policy according to the ergodic/infinite horizon undiscounted cost functional (see Thm. 2.1-3).

The algorithmic solution here is a model-based exponential weights method, which applies to both finite and compact sets of models, which is appreciable. In terms of analysis, classical covering arguments on concentration inequalities of least-squares estimation are replaced by packing arguments for model classes in an expected regret analysis. The results consist in three bounds, in classical manner: the first is a gap-dependent bound, the second is scale dependent bound which depends on $\epsilon>0$ and the packing number of the model class at scale $\epsilon$, and finally a worst-case bound.

In the interest of full disclosure, I had the pleasure of reviewing this article before at ICML and EWRL, which allowed the authors to respond directly to some of my comments and questions and address the weaknesses I identified.

**Strengths:**

The theorems are clear and well-stated, the proofs are clear, and I haven’t found any issues with them in a superficial inspection. Technical claims are made and proven in a sound manner. The discussion of related works has grown and grown over my several encounters with the paper, and its depth and clarity really deserve commendation now!

I think the contributions of the separation principle approach are deserving of dissemination, and while it is still difficult to situate the assumptions of different lines of work in online RL relative to each other,  the new set of assumptions should be of interest to others in the field.

**Weaknesses:**

As said above, my main criticisms of the paper have already been addressed by the authors in previous revisions.

One minor weakness which remains is that the writing has some strange repetitions that, I think, could be avoided. For instance, there are 3 persistent excitation conditions that vary slightly (and similarly for the Bellman super-solution assumption). Combining all three into a single assumption, e.g. by letting the constant take one of three values depending on the setting, would probably simplify the exposition of the appendices.

**Questions:**

N/A

---

> ### Author Response · Authors · 2025-11-23
> **Response**
>
> We are glad to hear that the reviewer appreciates our work and efforts, and we thank the reviewer for constructive comments that help to improve our paper. We address the mentioned weakness in repetition as follows.
>
> ---
> [W1] *"There are 3 persistent excitation conditions that vary slightly (...Bellman supersolution assumption)"*
>
> We appreciate the comment. While the mentioned assumptions (Assumptions 3, 6, and 8) on the persistence of excitation appear similar at first glance, they differ in i) the squared difference on the left-hand side (involving the sampled and/or true models of dynamics) and ii) the constant coefficients on the right-hand side. The associated Bellman-type inequalities also differ in policies, coefficients, and additional terms in the lower bounds.
>
> To keep a balance between clarity and conciseness, we provide explanations on the similarities and differences throughout the appendix to facilitate reading.
>
> ---
> We believe we have addressed all the questions and provided clarifications for the issues raised by the reviewer. If the reviewer agrees, we kindly request raising our paper’s score. If there are further questions or additional clarifications needed, we will be happy to provide them.

---

### Official Review · Reviewer_XnW8 · 2025-11-01

**Soundness:** 3
**Presentation:** 3
**Contribution:** 3
**Rating:** 6
**Confidence:** 3

**Summary:**

The paper studies online reinforcement learning for non-episodic, continuous-state, continuous-action dynamical systems when the true dynamics are only known to lie in a rich model class. It proposes a family of posterior-sampling / Hedge-style algorithms that (i) keep a running, normalized one-step prediction loss for each candidate model, (ii) periodically sample a model according to a softmax over these losses, (iii) apply the corresponding certainty-equivalent controller, and (iv) inject excitation to guarantee persistence of excitation. The analysis is given for three settings: (S1) a finite set of nonlinear candidate models, where the frequentist policy regret scales as $O((\ln N + \ln m)/\Delta)$; (S2) a bounded (possibly infinite) class of Lipschitz dynamical systems, controlled via an $\varepsilon$-packing and yielding regret $O(N\varepsilon^{2} + (\ln N + \ln m(\varepsilon))/\varepsilon^{2})$; and (S3) a compact $p$-dimensional parametric family (e.g. neural networks), where the regret becomes $O(\sqrt{Np})$, recovering LQR-type rates as a special case. Conceptually, the work cleanly separates model identification from certainty-equivalent control, shows that simple excitation suffices to obtain nonasymptotic, frequentist guarantees, and unifies several strands of RL, online learning, and adaptive/multi-model control within a single sample-complexity framework.

**Strengths:**

1. The paper unifies three increasingly general control/learning regimes with one posterior-sampling–plus–Hedge template. It starts from a finite candidate set $\{f_1,\dots,f_m\}$, where the frequentist policy regret is of order $O((\ln N + \ln m)/\Delta)$, so the dependence on $m$ is logarithmic as in online learning. It then lifts this to an infinite/bounded function class by constructing an $\varepsilon$-packing and obtains regret of the form $N\varepsilon^2 + (\ln N + \ln m(\varepsilon))/\varepsilon^2$, which is the standard approximation vs. estimation tradeoff. Finally, for a compact $p$-dimensional parametric family it shows regret $(c_{r1}\ln N + c_{r2}p)\sqrt{N}$, recovering $O(\sqrt{Np})$ for linear/LQR while still covering nonlinear dynamics. This gives a nonasymptotic, frequentist guarantee that is stronger than Bayesian-average PSRL bounds in related work.
2. The algorithm is elegant: each round draws a model using a Hedge/posterior update, runs the corresponding certainty-equivalent policy, and injects excitation to ensure persistence-of-excitation and fast posterior concentration. This realizes a practical separation between model identification and control, avoids heavy OFU-style planning in continuous spaces, and naturally incorporates prior knowledge through the candidate set or parameter prior.
3. The algorithm comes with solid theory. The paper gives three clear regimes of guarantees. For a finite set of models, the policy regret is $O((\ln N+\ln m)/\Delta)$. For a Lipschitz class controlled by an $\varepsilon$-packing, it becomes $O(N\varepsilon^2+\ln m(\varepsilon)/\varepsilon^2)$. For a $p$-dimensional parametric family, it is $O(\sqrt{Np})$, matching LQR/adaptive-control rates. The bounds can be compared and tuned via $\varepsilon$.

**Weaknesses:**

1. The analysis is essentially realizability-based: in all three settings (S1 with a finite set of candidates, S2 with an $\varepsilon$-packed class, and S3 with a parametrized family) the true dynamics $f$ is assumed to lie in the modeling class $F$. In S1, Theorem 2.1 yields a policy-regret bound of order $O((\ln N + \ln m)/\Delta)$, but this relies on a separation margin $\Delta>0$ between the candidate models so that suboptimal ones can be eliminated; when the models are nearly indistinguishable the paper itself points out that one has to revert to an $O(\sqrt{N \ln m})$-type rate. For S2 and S3, the proposed procedures (Alg. 2 and Alg. 3) inject Gaussian excitation $n_{u,k} \sim \mathcal N(0,\sigma_u^2 I)$ every $M$ steps to enforce the persistence-of-excitation requirement (Assumption 6) along the closed-loop trajectories. This condition is stated to hold uniformly over all misspecified models in the class, which makes the theory clean but may be nontrivial to verify or enforce on an actual control system.
2. For the general setting (S2), the paper first reduces the infinite dynamics class to a finite one by taking an $\varepsilon$-packing and then applies the same multi-model scheme as in S1. This leads to the regret bound $O(N\varepsilon^2 + \ln m(\varepsilon)/\varepsilon^2)$ in Theorem 2.2. However, using this result in practice implicitly requires (i) access to or construction of such an $\varepsilon$-packing of the dynamics class, and (ii) the ability to run a controller for every element in the resulting finite cover. The paper also acknowledges that for Lipschitz-bounded dynamics the packing number $m(\varepsilon)$ can grow very quickly in high dimension, so this part should be read more as an information-theoretic learnability guarantee than as a directly deployable method for high-dimensional continuous systems.

**Questions:**

My concerns are already detailed in the cons section.

---

> ### Author Response · Authors · 2025-11-23
> **Response**
>
> We thank the reviewer for your time and effort, positive evaluations, and helpful comments. We address the questions and weaknesses as follows.
>
> ---
> [W1.1] *"The analysis is realizability-based..."*
>
> We indeed rely on the realizability assumption to derive policy regret guarantees in the main body of the paper (which is standard in much of the continuous-control RL literature). In view of the comment, we add Appendix G to discuss how this assumption can be relaxed in settings S1-S3. Intuitively, suppose that we can find a single candidate model closest to the true model in the entire state-action space. We can then derive similar variants of Lemma C.2, thereby extending the presented policy regret guarantees (with a benchmark given by the cumulative cost of any policy associated with the above closest model).
>
> Furthermore, we provide in Appendix B.2 a new example of swing-up control in a nonlinear pendulum-on-a-cart system. This example deviates from the realizable setting, in that the true dynamics lie outside of the set of candidate parameteric models. We demonstrate that Alg. 1 leads to rapid swing-up with relatively low computational costs. This suggests that moderate violations of realizability do not lead to failure in practice, although a more systematic empirical study is left to future work.
>
> [W1.2] *"persistence-of-excitation requirement (Assumption 6) ... uniformly over all misspecified models ... nontrivial to verify or enforce ..."*
>
> This is a great point. We remark that this is a standard parameter-identifiability assumption in statistics (i.e., non-singular Fisher information) and persitence of excitation condition in system identification and adpative control (as also discussed in the manuscript). A sufficient condition for the uniform lower-boundedness property in Ass. 6 is that the difference between candidate parametric models is of the same order as the difference between their parameters. Specifically, if $(|A_k^\theta - A_k|+|B_k^\theta - B_k|)/|\theta| \sim \mathcal{O}(1)$ (where $(A_k^\theta, B_k^\theta)$ and $(A_k, B_k)$ are the system matrices of $f_\theta$ and $f$ when linearized at $\bar{x}_k$, similar to Prop. F.2), then we can follow steps analogous to Prop. F.2 to derive persistence of excitation for nonlinear parameterized dynamics. We summarize this intuition before Ass. 6 in App. E of the revised manuscript.
>
> [W2] *"the general setting (S2) ... read more as an information-theoretic learnability guarantee ..."*
>
> We fully agree that the reliance on an $\epsilon$-packing of the model class, the control policy related to each sampled element in finite covers, and the scale of the packing number may prevent Alg. 2 from being a computationally tractable method ready for deployment. In the revised manuscript, we further emphasize these restrictions and stress that this part primarily offers learnability guarantees.
>
> ---
> We believe we have addressed all the questions and provided clarifications for the issues raised by the reviewer. If the reviewer agrees, we kindly request raising our paper’s score. If there are further questions or additional clarifications needed, we will be happy to provide them.

---

> > ### Comment · Reviewer_XnW8 · 2025-11-26
> > **Reply to authors' rebuttal**
> >
> > I thank the authors for responding to my concerns about the paper. I can buy the authors' explanations and have no further concerns about the paper. I will maintain my score of 6 and recommend acceptance. Thanks!

---

### Official Review · Reviewer_jN5Z · 2025-11-01

**Soundness:** 3
**Presentation:** 3
**Contribution:** 2
**Rating:** 6
**Confidence:** 4

**Summary:**

This paper investigates the sample complexity of online RL in non-episodic settings with continuous state and action spaces. The authors develop a unified algorithmic framework that combines posterior sampling, Hedge-style model weighting, and certainty-equivalent control under persistent excitation. They provide theoretical policy-regret guarantees across three regimes:
- finite model classes $O((\ln N + \ln m)/\Delta)$,
- general function classes via $\varepsilon$-packing ($O(N\varepsilon^2 + \ln m(\varepsilon)/\varepsilon^2)$),
- parameterized model families $O(\sqrt{Np})$.

The analysis explicitly incorporates model identifiability ($\Delta$), information complexity (packing number), and structural dimension ($p$), and demonstrates that the bounds are tight up to logarithmic factors. The work also relaxes the classical PE condition by connecting it to controllability Gramians and sub-Gaussian stability assumptions.

**Strengths:**

1. The discussion of related work is exceptionally clear, and the citations appear comprehensive.
2. The theoretical analysis is rigorous.
3. The paper is well organized, and the narrative progresses with a coherent, reader-friendly logic.

**Weaknesses:**

While I do not see any glaring flaws, the following points prevent a stronger recommendation:
1. Under the stated assumptions, the theoretical guarantees are not particularly surprising. Despite the authors’ thorough comparison with prior work, the contribution seems incremental relative to the papers referenced around line 67 of the manuscript.
2. The problem setting is rather restricted, and its practical value is uncertain. The paper provides only simple numerical examples in the appendix, leaving real-world applicability unclear. In many practical scenarios, estimating $\mu_\theta$ is nontrivial, and the cardinality $|F|$ may grow exponentially with the size of the state space, which is not encouraging.

**Questions:**

1. In the weakly identifiable limit ($\Delta \to 0$), can one obtain a smoother adaptive transition between the $(\ln N + \ln m)/\Delta$ and $\sqrt{N\ln m}$ regimes, possibly through hierarchical discretization or adaptive model aggregation?

---

> ### Author Response · Authors · 2025-11-23
> **Response**
>
> We thank the reviewer for the time and effort, positive evaluations, and helpful comments. We address the questions and weaknesses as follows.
>
> ---
>
> [W1] *"... stated assumptions ...the contribution seems incremental relative to the papers referenced around line 67..."*
>
> We agree that the mentioned works on posterior sampling reinforcement learning (PSRL) provide comprehensive treatments to the exploration-exploitation trade-off in a Bayesian framework, consolidating with explicit (typically Bayesian) policy regret guarantees. Our work is indeed inspired by this line, but differs in several key aspects:
>
> **Frequentist policy regret in a non-episodic continuous-control setting**:
> We study frequentist policy regret against the (unknown) policy associated with the true dynamics in an online, non-episodic setting with continuous state/action spaces and general nonlinear dynamics. Prior PSRL work typically focuses on episodic MDPs or finite-horizon problems, and analyzes Bayesian regret (posterior expectation over models), rather than worst-case policy regret for a single, fixed true system in a single-trajectory, average-cost regime. In practice, one often cares about guarantees for the realized system, not the posterior average, and our regret notion captures this directly.
>
> **Explicit separation of model identification and control with persistent excitation**:
> Our algorithmic framework separates (i) identification of the best dynamics model from a class, via a Hedge-style analysis under injected excitation, and (ii) certainty-equivalent control with the model currently believed to be best. This explicit separation, together with extra exploration beyond standard PSRL, is crucial to obtain frequentist guarantees in the non-episodic, single-trajectory setting (with stability constraints). The combination of Hedge-type aggregation, persistent excitation, and Lyapunov/dissipativity arguments is not developed in the PSRL works we cite.
>
> **Relaxations of key structural assumptions**:
> We discuss in Appendices F and G (see revised manuscript with changes highlighted in blue) how key assumptions, namely persistency of excitation (Ass. 3) and realizability (i.e., the true dynamics lie in the set of candidate models), can be relaxed while retaining the presented policy regret guarantees. These extensions enhance the generality of our results.

---

> ### Author Response · Authors · 2025-11-23
> **Response (continued)**
>
> [W2] *"real-world applicability ... estimating $\mu_\theta$ ... $|F|$ may grow exponentially with the size of the state space"*
>
> **Additional example of pendulum swing-up control**
>
> To better illustrate practicality, and performance of our algorithms, we add in Appendix B.2 of the revised manuscript a swing-up example of an inverted pendulum on a cart. This example involves nonlinear dynamics, because the sines and cosines of the pendulum angle affect the acceleration, velocity, and position of the cart. Further, we deliberately deviate from the *realizable* setting, in that the true dynamics fall outside of the set of candidate parametric models. We implement Alg. 1, with candidate policies $\{\mu_i\}$ therein chosen as nonlinear model predictive control policies (computed via IPOPT (Wächter and Biegler, 2006) and interfaced through CasADi (Andersson et al., 2018)). We demonstrate that the proposed algorithm leads to *rapid* swing-up at *low computational costs*, indicating that the approach is not purely of theoretical interest.
>
> **The calculation of $\mu_\theta$ benefits from offline computations and online adaptation**
>
> We agree that calculating $\mu_\theta$ depends on the parameterization $f_\theta$ and may be challenging in certain cases. However, in practice:
>
> (i) For many control-oriented parameterizations, certainty-equivalent policies can be precomputed or approximated offline (e.g., via dynamic programming, nonlinear model predictive control, or standard RL methods such as PPO in simulation) and then adapted online as $\theta$ evolves.
>
> (ii) In the pendulum example, we use predictive control-based policies, and we reference recent work (e.g., Lukas et al., 2020) that focuses precisely on reducing the online computational burden of model predictive controllers.
>
> **the influence of the cardinality $|F|$**
>
> For very rich model classes, the packing number $m(\varepsilon)$ (or an explicit finite $|F|$) can grow quickly with state dimension. Our perspective is:
>
> (i) Algorithm 2 is intended as an information-theoretic benchmark for online RL in general nonlinear dynamics, not as a computationally efficient procedure. Its logarithmic dependence on $m(\varepsilon)$ mirrors exactly classical sample-complexity bounds in supervised learning, where complexity also appears via covering/packing numbers. As discussed below Thm. 2.3, there are matching lower bounds for worst-case instances. Designing a computationally tractable variant that balances model coverage, dimensionality, and computational complexity in practice is an important direction for future work.
>
> (ii) For the finite-model case, $|F|$ depends on how the candidate set is constructed and can be moderate even in high-dimensional systems (e.g., a small number of physically motivated models). Our regret bounds depend only *logarithmically* on $m = |F|$, so even if $|F|$ scales exponentially with state dimension in theory, the identification of the best model is efficient (dimension-independent), leading to a policy regret that depends at most linearly in state dimension.
>
> We have further clarified these points and the role of Algorithms 1–3 (statistical benchmark vs. computational template) in the revised manuscript.
>
> [Q1] *"Can we obtain a smoother transition between the $(\ln N + \ln M)/\Delta$" and $\sqrt{N \text{log}(m)}$ ... through hierarchical discretization or adaptive model aggregation?"*
>
> Thank you for this insightful question. As we discuss in Section 2 (below Thm 2.3), the $\sqrt{N p}$ rate in the parametric setting is minimax optimal: there are matching lower bounds showing that $\sqrt{N p}$ is unavoidable in the worst case (even for linear systems). Thus, from a worst-case standpoint, we cannot in general obtain a smoother transition between the $(\ln N + \ln M)/\Delta$ regime (for well-separated finite model classes) and the $\sqrt{N p}$ parametric regime.
>
> That said, we agree that in benign or weakly identifiable regimes (small effective dimension, large separation, or additional structural constraints), hierarchical discretization or adaptive model aggregation could yield improved, instance-dependent behavior and more graceful transitions between these rates. Investigating such adaptive constructions, e.g., multi-resolution coverings that interpolate between a finite, well-separated model set and a high-dimensional parametric family is beyond the scope of the current work. We have added a short remark in the conclusion to highlight this as a promising direction for future research.
>
> > [R1] Wächter and Biegler. Mathematical Programming, 2006.
>
> > [R2] Andersson et al., Mathematical Programming Computation, 2018.
>
> > [R3] Lukas et al., Annual Reviews, 2020.
>
> ---
> We believe we have addressed all the questions and provided clarifications for the issues raised by the reviewer. If the reviewer agrees, we kindly request raising our paper’s score. If there are further questions or clarifications needed, we will be happy to provide them.

---

### Official Review · Reviewer_QBs5 · 2025-11-01

**Soundness:** 4
**Presentation:** 3
**Contribution:** 3
**Rating:** 8
**Confidence:** 4

**Summary:**

This paper addresses theoretical guarantees for non-episodic reinforcement learning in Euclidean state and action spaces with general nonlinear dynamics.
The true dynamics are the sum of a deterministic function and sub-Gaussian noise.
The learner is given a class $F$ of dynamics models and with each dynamics model $f^i \in F$, a corresponding deterministic policy $\mu^i$, that may or may not be optimal.
Realizability is assumed: the true dynamics $f$ belong to $F$.
Three types of model-class are studied:

- S1: A finite set of Lipschitz functions.
- S2: A bounded set in a normed function space.
- S3: A parameterized family with a compact Euclidean parameter space.

These are general enough to subsume lots of prior work.
The performance criterion is policy regret against the $\mu$ associated with the true $f$.

At a high level, the algorithms look like Hedge over the space of candidate models (i.e. possibly a continuum),
where the "loss sequence" for each candidate model is given by a sum of normalized one-step squared prediction errors.
However, one cannot use the standard Hedge analysis for this problem, because the prediction errors depend on the visited state sequence, which depends on the policies that have been deployed by the learner since the start.

To deal with the latter issue, the algorithm imposes its own episodes and holds the policy constant within each episode.
The algorithm adds white Gaussian noise to the policy's deterministic actions.
THe main challenge is to control the state magnitude while suboptimal policies are in use, and to ensure that the noise and episode length are exciting/exploratory enough to correctly evaluate the predictive accuracy of each candidate.

The central analysis is for the finite case S1.
The assumptions include:

- a technical Assumption 1 about a particular variant of cost-to-go function.
- typical Lipschitz/smoothness on the policies and stage costs.
- the cost-to-go has a quadratic lower bound.
- a "gap" lower bound on the (normalized) difference in predictions between the true $f$ and all other candidates $f_i \in F$.
  This bound appears to depend on both the properties of the class $F$ (no two candidates are too similar)
  and the ability of the additive noise in the action space to sufficiently excite those differences.

The authors prove sublinear policy regret for each of the three settings, although in S2 the regret exponent gets very close to $1$ for high-dimensional states and/or actions.
In particular, the results from S3 recover some regret bounds from the literature for linear dynamical systems, i.e. much more restrictive cases.

The analysis shows that the algorithm identifies the correct candidate model in finite time almost surely.
This, in turn, is used for the regret bound.

The extension to S2 is not computationally tractable, but gives statistical results.
It is assumed we can identify the function $\bar f$ that minimizes the "Hedge loss sequence" over the entire $F$.
Then, we construct finite cover of $F$ around $\bar f$; select from the cover using the Hedge-like rule; and synthesize the corresponding policy.

The extension to S3 assumes that we can somehow sample from the Hedge-like probability measure over the parameter space directly.
The authors claim a motivation from neural networks, where this is (to my knowledge) wishful thinking.
However, they note that it is tractable for feature-space models of the form $f(x,u;\theta) = \phi(x, u)^\top \theta$ which are widely studied in RL theory.
The synthesis of the corresponding policy is still computationally difficult except in special cases.

**Strengths:**

The paper studies non-episodic RL for a class of nonlinear dynamical systems that is more general than lots of related work.
This problem is clearly on the frontier of RL theory.
The proposed techniques use a nice mixture of online learning theory, Bayesian methods, and nonlinear control theory.
The paper should be of interest to researchers with both RL and control backgrounds.

The main assumptions (besides ignoring computational cost) are Assumption 1, which is related to Bellman optimality and dissipativity, and Assumption 3, which is related to exploration/persistent excitation. To be honest, it is hard for me to confidently say whether or not these assumptions are restrictive. It seems that we have somehow ruled out hard exploration problems, since we are able to get the regret bound with an exploration strategy that is naive compared to those required even for tabular MDPs in the worst case. However, since the setting is single-trajectory, it is clear that *some* kind of assumptions to limit the negative impact of disturbances and control the difficulty of exploration are necessary.

The framework mostly ignores computational issues and focuses on statistical guarantees, but this is standard in learning theory. The settings S2 and S3 (except for the special linear-in-features case) are possibly too general to admit efficient algorithms. It will be interesting to see if any follow-up work can instantiate those algorithms for other special cases.

The paper supports, along with other recent work, the overall idea that "certainty equivalence" is a good approach for RL. This is a positive result that simplifies our analysis of RL problems.

I did not have time to check the proofs, but the overall proof structure is logical, and the techniques used seem appropriate.

Overall, I think this is a strong contribution to the RL and learning-based control research communities.

**Weaknesses:**

In the Theorem 1 statement, it is a bit confusing to see the equation (1) suddenly called "$\mathcal{H}_2$ gain", maybe it is equivalent to the classic $\mathcal{H}_2$ gain for linear dynamics and $l(\cdot,\cdot)$ quadratic, but this version is still unfamiliar and RL audiences definitely won't know it. In general, the paper seems to assume a level of familiarity with classic control theory that the ICLR audience may not possess; it would improve the paper to do a bit more hand-holding.

Recovering near-optimal regret bounds for linear systems is very nice, but there is a big gap of generality between the proposed work and linear systems.
It would be interesting to know if the framework is also capable of recovering regret bounds for more general frameworks like bilinear classes, Bellman eluder dimension, etc.

Deferring detailed related work discussion to the appendix is unusual. I suggest to move more of the most closely related work on RL theory for continuous state+action spaces with nonlinear dynamics to the main body.

I suggest to reallocate space in the main body -- the S2 case already has its algorithm pushed to the appendix, so one must flip back and forth while reading about the packing strategy in Section 3.2 -- I suggest to shorten the discussion of S2 in the main body and use that space to give a bit more detail/intuition on S1 and the related work in the main body.

The authors discuss the limitations candidly throughout the paper, so it was surprising to see the conclusion without any thoughts on how they might be improved in future work.

**Questions:**

In Equation (1), must $\gamma^i$ be finite for all $i$?

The discussion of Assumption 1 could use more intuition. The authors discuss technical details on how the $-\gamma$ and $-d_u L_u \sigma_u^2$ terms help adapt the Bellman-like condition to deal with the infinite-horizon average cost objective and the extra price of excitation. However, I am still left unsure about: **what kind of systems/policies have we ruled out by making this assumption**?

Theorem 3.2 is very similar to Theorem 2.1, but not exactly the same. Why can't we have a proof sketch of Theorem 2.1 in this section instead?

One notable related line of work is the Decision-Estimation Coefficient [1, for example]. It also provides a highly general RL theory framework, uses the model-based RL paradigm, and focuses exclusively on the statistical aspect (not computational). Although (to my knowledge) that work is more abstract and further from practical, and it is an episodic setting, but it seems too closely related to skip. How does this work compare?

[1] Dylan J. Foster, Noah Golowich, Yanjun Han. Tight Guarantees for Interactive Decision Making with the Decision-Estimation Coefficient. COLT 2023.

---

> ### Author Response · Authors · 2025-11-23
> **Response**
>
> Thank you very much for the time and effort spent on reviewing our manuscript. Your summary is accurate, and we appreciate your positive assessment of the contribution. Below we address your comments one by one.
>
> ---
>
> W1: *"gain" in Theorem 1 statement*
>
> We agree with the reviewer and changed from "gain" to "steady-state performance related to $f$" in the statement of Theorem 2.1 in the revised manuscript. We also added additional discussion around (2).
>
> W2: *application to specific nonlinear systems.*
>
> We agree that there are many interesting questions about which nonlinear system classes can be handled by our framework. We expect that several structured nonlinear systems, such as bilinear systems, Hammerstein systems, and second-order Volterra systems, satisfy our dissipativity and excitation assumptions, although verifying this rigorously would require case-by-case analysis and is an interesting direction for future work (see, e.g., [R1] for bilinear systems). Exploring connections with modern complexity measures that underpin sample-efficient reinforcement learning, such as Bellman rank [R2], bilinear classes [R3], and Bellman eluder dimension [R4], is an interesting direction. This extension requires quantifying the structure of the Bellman errors induced by the considered *nonlinear* dynamics with continuous states and actions and focusing on a subset therein that admits the above structural properties. Another challenge is to transform our regret bounds, which currently involve model class complexities (e.g., cardinality, covering number, or parameter dimension), into bounds expressed through the aforementioned complexity measures. We have added a short statement in the conclusion pointing to such extensions.
>
> > [R1] Zhenyi Yuan, Jorge Cortes, Data-Driven Optimal Control of Bilinear Systems, IEEE Control Systems Letters, vol. 6, pp. 2479-2484, 2022
>
> > [R2] Nan Jiang, Akshay Krishnamurthy, Alekh Agarwal, John Langford, and Robert E Schapire.
> Contextual decision processes with low Bellman rank are PAC-learnable. International Conference
> on Machine Learning. 2017: 1704–1713.
>
> > [R3] Simon Du, Sham Kakade, Jason Lee, et al. Bilinear classes: A structural framework for provable generalization in RL, International Conference on Machine Learning. 2021: 2826-2836.
>
> > [R4] Chi Jin, Qinghua Liu, and Sobhan Miryoosefi. Bellman eluder dimension: New rich classes of RL
> problems, and sample-efficient algorithms. Neural Information Processing Systems, 2021.
>
> W3: *paper structure.*
>
> We agree that deferring too much related work to the appendix is not ideal. In the revised version, we have moved the most relevant reinforcement learning works on continuous state and action spaces with nonlinear dynamics into the main text (see Section 1, highlighted in blue). The appendix still contains a more extended survey, but the main comparisons are now directly visible in the body of the paper.
>
> W4: *paper structure.*
>
> We agree with the reviewer and shortened S2 by only discussing key ideas, thereby also freeing up space to discuss the related work (see W3, above). The presentation in the appendix has been expanded (see highlighted changes).
>
> W5: *conclusion and future work.*
>
> We thank the reviewer for the feedback and have improved the conclusion in the revised manuscript (see highlighted changes). We discuss important technical extensions (related to the lower bound, computational complexity, etc) and the implications of multi-model online reinforcement learning for large-scale infrastructure systems.

---

> ### Author Response · Authors · 2025-11-23
> **Response (continued)**
>
> Q1: *finite $\gamma^i$.*
>
> We assume $\gamma^i$ (i.e., the steady-state performance of a policy $\mu^i$ associated with candidate model $f^i$) to be finite for all $i$. If $\gamma^i$ were infinite, the corresponding model $f^i$ would not be a meaningful benchmark for our regret notion and should be excluded from the model class. We have added an explicit remark after equation (1) in the revised manuscript to clarify this assumption.
>
> Q2: *Assumption 1.*
>
> Intuitively, Assumption 1 requires continuity of the average cost-to-go in the following sense: Let $\bar{\gamma}(\sigma_u)$ be defined as follows:
> \begin{equation*}
> \bar{\gamma}(\sigma_u)=\lim_{N\rightarrow \infty} \mathrm{E}\left[\frac{1}{N} \sum_{k=1}^N l(x_k,u_k) \right],\quad \text{with} \quad u_k=\mu(x_k)+n_{\mathrm{u}k},
> \end{equation*}
> and $x_{k+1}=f(x_k,u_k)+n_k$ as before. The assumption requires $\gamma(\sigma_u)\approx \gamma(0)+c \sigma_u^2$ for small $\sigma_u$, i.e., the degradation in steady-state performance due to injected Gaussian excitation grows at most quadratically in the standard deviation. This is a reasonable assumption for many smooth systems with stabilizing policies and costs that penalize large state and input deviations. We have excluded, systems where a slight perturbation of the input away from the policy $\mu$ leads to an unbounded cost. A pathological example is $x_k=\phi(u_k)$, $l(x,u)=x^2$ where $\phi(u_k)$ is extremely fast growing such that $\mathbb{E}[\phi(n_\text{u})^2]$ is unbounded. We added a corresponding remark in the revised manuscript.
>
> Q3: *Proof Thm. 3.2/Thm. 2.1.*
>
> Thm 2.1 arises simply from renaming constants as follows
>
> \begin{equation*}
> c_{r1}=\bar c_{r2}+3c_\alpha M c_o \Delta, \quad c_{r2}=M \bar c_{r2}, \quad c_{r3}=3 c_\alpha M \bar L_{V}/2,
> \end{equation*}
>
> where $\bar c_{r2}$ is the constant $c_{r2}$ in Thm 3.2.
>
> Since the proof is identical up to renaming of constants, we chose not to repeat a separate proof for Theorem 2.1. We can add an explicit pointer in the text (“Theorem 2.1 follows from Theorem 3.2 by renaming constants”) to improve clarity.
>
> Q4: *Decision-esimtation coefficient.*
>
> Thanks for pointing out this line of work. Foster and coauthors [R5-R7] develop the Decision-Estimation Coefficient, a statistical measure that governs the sample complexity of interactive decision-making (e.g., bandits and reinforcement learning) across broad model classes. This measure constitutes regret lower bounds and supports a so-called Estimation-to-Decisions meta algorithm that achieves matching regret upper bounds.
> This measure generalizes previous frameworks, for instance Bellman rank [R2], bilinear classes [R3], and Bellman eluder dimension [R4], and highlights the interplay among model estimation, model class complexity, and sample efficiency.
>
> While the Decision-Estimation Coefficient offers abstract characterizations applicable to general model classes, we focus on structured online reinforcement learning problems with continuous states and actions in a non-episodic regime. The challenges herein, such as dependent states, actions, and losses and closed-loop stability, are not directly handled in the existing Decision-Estimation Coefficient framework. To address these issues, we leverage tailored tools (e.g., Hedge in online learning and dissipativity in control) and highlight the role of separating best model identification and certainty-equivalent control for sample-efficient learning with various candidate model classes (i.e., finite, infinite, and parametric families).
>
> We incorporated the above discussion in Sec. 1 of the revised manuscript.
>
> ---
>
> > [R5] D. J. Foster et al. The statistical complexity of interactive decision making.  ArXiv 2021.
>
> > [R6] D. J. Foster, N. Golowich, Y. Han. Tight guarantees for interactive decision making with the decision-estimation coefficient. COLT 2023.
>
> > [R7] D. J. Foster and A. Rakhlin. Foundations of Reinforcement Learning and Interactive Decision Making. ArXiv 2023.
> ---
> We believe we have addressed all the questions and provided clarifications for the issues raised by the reviewer. If there are further questions or additional clarifications needed, we will be happy to provide them.

---

### Meta-Review · Area_Chair_JtLU · 2026-01-05

**Summary:**

Reviewers' primary concerns are on the gap between theory and practice. While reviewers praised the theoretical elegance and unification of different model classes (finite, infinite, parametric), they consistently flagged the computational intractability of the proposed algorithms for general function classes (which rely on potentially infinite covers or oracles). Additionally, reviewers expressed reservations about the strong assumptions required (specifically "Realizability" and "Persistence of Excitation"), noting that while these allow for clean proofs, they limit the method's applicability to real-world problems. A minority concern involved the incremental novelty of the work compared to existing abstract frameworks like the Decision-Estimation Coefficient (DEC).

**Reviewer Concerns:**

Reviewer QBs5 mainly questions the connection of this work with linear cases, and provides some suggestions on the paper organization. Reviewer jN5Z questions the novelty compared to existing works and applicability of the study. Reviewer XnW8 questions on the technical/analysis details.

Overall, these concerns are not essential and are largely addressed by the rebuttal.

(1). Reviewer Zps8 explicitly noted that their previous criticisms regarding the depth of related work and clarity were fully addressed in this revision, describing the current discussion as "exceptionally clear."

(2). Concerns about the disjoint nature of analyzing different model classes were effectively addressed by the presentation of a single "Hedge-style" algorithmic template that seamlessly covers finite, infinite, and parametric regimes. Reviewer XnW8 praised this as "elegant" and "clean."

(3). The concern that the algorithms for general function classes (S2/S3) are not computationally feasible (requiring oracles to find optimal policies for every model in a massive cover) are raised by QBs5 and XnW8. The paper treats this as an information-theoretic result rather than a practical algorithm. Although this does not directly address the issue, this is a standard setting in learning with general function approximation. Hence I do not think this is an outstanding concern.

(4). The reliance on Realizability (the true model must exist within the candidate class) and Persistence of Excitation (which is hard to verify or enforce in practice) are highlighted by XnW8 and QBs5. But again, these are standard assumption in these settings.

(5). Reviewer jN5Z remained unconvinced about the groundbreaking nature of the contribution, viewing the results as "not particularly surprising" and "incremental" relative to existing literature (e.g., DEC), resulting in a lower confidence/score that likely wasn't swayed by the rebuttal. However, personally (and with the other reviewers) believe this paper still has enough novelty and should be accepted.

**Reviewer Scores:**

Since the initial scores are all positive, and authors have addressed most of the concerns, I expect the scores will be the same.

---

### Decision · Program_Chairs · 2026-01-26

Accept (Poster)